# CREATIVE ROBOT TOOL USE WITH LARGE LANGUAGE MODELS

## ABSTRACT

Tool use is a hallmark of advanced intelligence, exemplified in both animal behavior and robotic capabilities. This paper investigates the feasibility of imbuing robots with the ability to creatively use tools in tasks that involve implicit physical constraints and long-term planning. Leveraging Large Language Models (LLMs), we develop RoboTool, a system that accepts natural language instructions and outputs executable code for controlling robots in both simulated and real-world environments. RoboTool incorporates four pivotal components: (i) an "Analyzer" that interprets natural language to discern key task-related concepts, (ii) a "Planner" that generates comprehensive strategies based on the language input and key concepts, (iii) a "Calculator" that computes parameters for each skill, and (iv) a "Coder" that translates these plans into executable Python code. Our results show that RoboTool can not only comprehend implicit physical constraints and environmental factors but also demonstrate creative tool use. Unlike traditional Task and Motion Planning (TAMP) methods that rely on explicit optimization and are confined to formal logic, our LLM-based system offers a more flexible, efficient, and user-friendly solution for complex robotics tasks. Through extensive experiments, we validate that RoboTool is proficient in handling tasks that would otherwise be infeasible without the creative use of tools, thereby expanding the capabilities of robotic systems. Demos are available on our project page.[1]

## 1 INTRODUCTION

Tool use is an important hallmark of advanced intelligence. Some animals can use tools to achieve goals that are infeasible without tools. For example, Koehler's apes stacked crates together to reach a high-hanging banana bunch (Kohler, 2018), or the crab-eating macaques used stone tools to open nuts and bivalves (Gumert et al., 2009). Beyond using tools for their intended purpose and following established procedures, using tools in creative and unconventional ways provides more flexible solutions, albeit presents far more challenges in cognitive ability. In robotics, creative tool use (Fitzgerald et al., 2021) is also a crucial yet very demanding capability because it necessitates the all-around ability to predict the outcome of an action, reason what tools to use, and plan how to use them. In this work, we want to explore the question, *can we enable such creative tool-use capability in robots?* We identify that creative robot tool use solves a complex long-horizon planning task with implicit constraints of environment and robot capacity. For example, "grasp a milk carton" while the milk carton's location is out of the robotic arm's workspace or "walk to the other sofa" while there exists a gap in the way that exceeds the quadrupedal robot's walking capability.

Task and motion planning (TAMP) is a common framework for solving such long-horizon planning tasks. It combines low-level continuous motion planning in classic robotics and high-level discrete task planning to solve complex planning tasks (Pezzato et al., 2023). Existing literature shows that it can handle tool use in a static environment with optimization-based approaches such as logic-geometric programming (Toussaint et al., 2018). However, this optimization approach generally requires a long computation time for tasks with many objects and task planning steps due to the increasing search space (Garrett et al., 2021). In addition, classical TAMP methods are limited to the family of tasks that can be expressed in formal logic and symbolic representation, making them not user-friendly for non-experts (Chen et al., 2023; Lin et al., 2023b).

---

[1]Project Page: https://creative-robotool.github.io/.

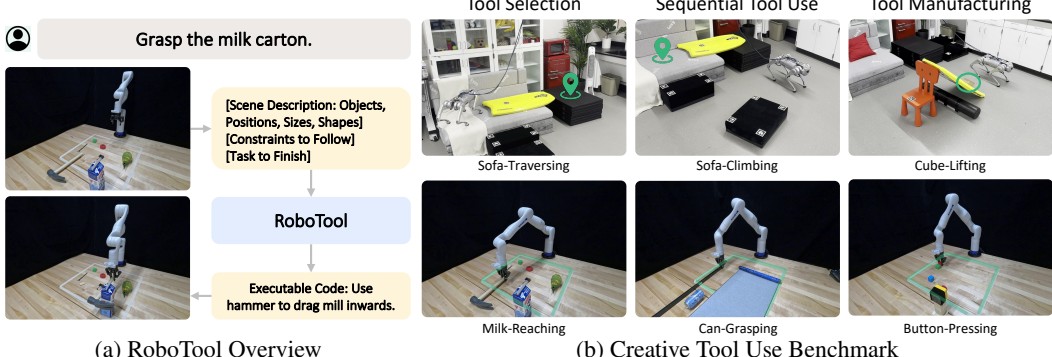

(a) RoboTool Overview       (b) Creative Tool Use Benchmark

Figure 1: (a) Creative robot tool use with Large Language Models (RoboTool). RoboTool takes natural language descriptions as input, including the scene descriptions, environment- and embodiment-related constraints, and tasks. (b) We design a creative tool-use benchmark based on a quadrupedal robot and a robotic arm, including 6 challenging tasks that symbols three types of creative tool-use behaviors.

Recently, large language models (LLMs) have been shown to encode vast knowledge beneficial to robotics tasks in reasoning, planning, and acting (Brohan et al., 2023; Huang et al., 2023b; Yu et al., 2023). TAMP methods with LLMs are able to bypass the computation burden of the explicit optimization process in classical TAMP. Prior works show that LLMs can adeptly dissect tasks given either clear or ambiguous language descriptions and instructions. Robots powered by LLMs also demonstrate notable compositional generalization in TAMP (Huang et al., 2022a; Ahn et al., 2022). However, it is still unclear how to use LLMs to solve more complex tasks that require reasoning with implicit constraints imposed by the robot's embodiment and its surrounding physical world.

In this work, we are interested in solving language-instructed long-horizon robotics tasks with implicitly activated physical constraints (Fig. 1). By providing LLMs with adequate numerical semantic information in natural language, we observe that LLMs can identify the activated constraints induced by the spatial layout of objects in the scene and the robot's embodiment limits, suggesting that LLMs may maintain knowledge and reasoning capability about the 3D physical world. For example, in the Sofa-Traversing example, the LLM can identify that the key concept affecting the plan's feasibility is the gap width between two sofas, although there is no prior information about the existence of the "gap" concept in the provided language instruction. Furthermore, our comprehensive tests reveal that LLMs are not only adept at employing tools to transform otherwise unfeasible tasks into feasible ones but also display creativity in using tools beyond their conventional functions, based on their material, shape, and geometric features. Again, in the Sofa-Traversing example, the LLM could use the surfboard next to the quadrupedal robot as a bridge to walk across the gap.

To solve the aforementioned problem, we introduce RoboTool, a creative robot tool user built on LLMs, which uses tools beyond their standard affordances. RoboTool accepts natural language instructions comprising textual and numerical information about the environment, robot embodiments, and constraints to follow. RoboTool produces code that invokes robot's parameterized low-level skills to control both simulated and physical robots. RoboTool consists of four central components, with each handling one functionality, as depicted in Fig. 2: (i) *Analyzer*, which processes the natural language input to identify key concepts that could impact the task's feasibility, (ii) *Planner*, which receives both the original language input and the identified key concepts to formulate a comprehensive strategy for completing the task, (iii) *Calculator*, which is responsible for determining the parameters, such as the target positions required for each parameterized skill, and (iv) *Coder*, which converts the comprehensive plan and parameters into executable code. All of these components are constructed using GPT-4.

Our key contributions are in three folds:

- We introduce RoboTool, a creative robot tool user based on pretrained LLMs, that can solve long-horizon hybrid discrete-continuous planning problems with environment- and embodiment-related constraints in a zero-shot manfner.

- We provide an evaluation benchmark (Fig. 1b) to test various aspects of creative tool-use capability, including tool selection, sequential tool use, and tool manufacturing, across two widely used embodiments: the robotic arm and the quadrupedal robot.

- Simulation and real-world experiments demonstrate that RoboTool solves tasks unachievable without creative tool use and outperforms baselines by a large margin in terms of success rates.

## 2 RELATED WORKS

**Language Models for Task and Motion Planning (TAMP).** TAMP (Garrett et al., 2021) has been integrated with LLMs for building intelligent robots. Most of the literature built upon hierarchical planning (Garrett et al., 2020; 2021; Kaelbling & Lozano-Pérez, 2011), where LLMs only provide a high-level plan that invokes human-engineered control primitives or motion planners (Ahn et al., 2022; Huang et al., 2022a;b; Ren et al., 2023a; Chen et al., 2023; Ding et al., 2023; Silver et al., 2023; Liu et al., 2023; Xie et al., 2023). In this work, we follow the hierarchical planning setting and aim to develop an LLM-based planner to solve tasks with constraints that require creative tool-use behaviors. One challenge of using LLMs as a planner is to ground it with real-world interactions. SayCan (Ahn et al., 2022), Grounded Decoding (Huang et al., 2023c) and Text2Motion (Lin et al., 2023a) grounded an LLM planner with a real-world affordance function (either skill-, object- or environment-related) to propose feasible and appropriate plans. Developing these affordance functions requires extra training from massive offline data or domain-specific knowledge. In contrast, we rely entirely on LLM's capability of deriving the affordance from the language input and do not require separate pretrained affordance functions.

Existing works integrating LLM and TAMP output the plan either in the format of natural language, PDDL language, or code scripts. With the focus on robotics applications, one natural interface to call the low-level skills is code generated by LLMs. Code-as-Policies (Liang et al., 2023) and ProgPrompt (Singh et al., 2023) showed that LLMs exhibit spatial-geometric reasoning and assign precise values to ambiguous descriptions as well as writing snippets of logical Python code. In the multi-modal robotics, Instruct2Act (Huang et al., 2023a) and VoxPoser Huang et al. (2023b) also generate code to incorporate perception, planning, and action. Following these works, we propose to use a standalone LLM module to generate codes.

**Robot Tool Use.** Tool use enables robots to solve problems that they were unable to without tools. There is a long history of interest in robotics literature that focuses on manipulating tools to finish designated tasks, such as furniture polishing (Nagata et al., 2001), nut fastening (Pfeiffer et al., 2017), playing table tennis (Muelling et al., 2010), using chopsticks Ke et al. (2021) etc. Despite these successful attempts, they focus on generating actions for specific tools and do not study the causal effect of tools and their interaction with other objects. Therefore, they cannot deal with novel objects or tasks and generally lack improvisational capability. To interact with tools in a more intelligent way, Sinapov & Stoytchev (2008) and Levihn & Stilman (2014) conducted early attempts to study the effects of different tools and mechanisms. Wicaksono & Sammut (2016) developed a system to learn a simple tool from a demonstration of another agent employing a similar tool by generating and testing hypotheses, represented by Horn clauses, about what tool features are important. Xie et al. (2019) and Fang et al. (2020) trained deep neural networks from diverse self-supervised data of tool use to either predict the effects of tools or generate the action directly.

Related to our method, Toussaint et al. (2018) formulated a Logic-Geometric Program to solve physical puzzles with sequential tool use, such as using a hook to get another longer hook to reach for a target ball. Ren et al. (2023b) utilized LLMs to transform task and tool features in text form as latent representations and concatenate them with vision input to achieve faster adaptation and generalization via meta-learning. More recently, RT2 (Brohan et al., 2023) performed multi-stage semantic reasoning to, for example, decide the rock could be used as an improvised hammer. Unlike most of the existing robotics literature, our method leverages LLM's massive prior knowledge about object affordance and impressive planning capability to propose creative solutions to different kinds of physical puzzles. These solutions require highly complex reasoning and planning capability.

## 3 METHODOLOGY

We are interested in enabling robots to solve complex long-horizon tasks with multiple environment- and embodiment-related constraints, that require robot's *creative tool-use* capability to solve the tasks. In this section, we first posit our problem as a hybrid discrete-continuous planning problem in Sec. 3.1. We then introduce our proposed method, RoboTool, in Sec. 3.2, which is a creative robot tool user built on LLMs and can solve complex task planning problems in a zero-shot manner.

### 3.1 PROBLEM FORMULATION

With a focus on robotic applications, we aim to solve a hybrid discrete-continuous planning problem with multiple constraints based on a natural language description. The provided description contains

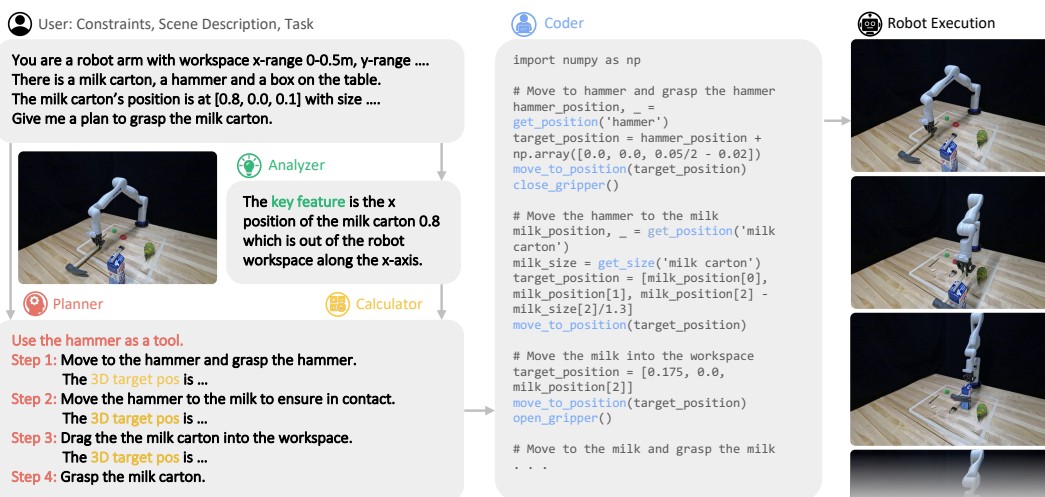

Figure 2: Overview of our proposed RoboTool, which is a creative robot tool user consisting of four key components including *Analyzer*, *Planner*, *Calculator* and *Coder*.

words and numerical values to depict the environments, tasks, and constraints. Assuming the robot is equipped with a repertoire of parameterized skills, we seek a hierarchical solution to generate a plan composed of a sequence of provided skills and a sequence of parameters for the corresponding skills. As code serves as a general interface to send commands to robots, our method will output executable robot policy code that sequentially calls skills with the parameters according to the plan, to complete the task. Solving such problems typically involves interacting with different objects in the scene to satisfy the environment and embodiment constraints. This symbolizes *the tool-use behavior*, which is a cornerstone of intelligence.

**Language Description as Inputs.** We define the environment layout space as $\mathcal{Q}$ and the initial environment configuration as $q_0 \in \mathcal{Q}$. The API $F_q$ helps parse $q_0$ into an environment language description $L_{\mathcal{Q}} = F_q(q_0)$, which includes the spatial layouts of objects, such as "there is a hammer on the table" and "the robot is on the ground," as well as each object's positions, sizes, and physical properties. We denote the constraint set as $\mathcal{C} = \mathcal{C}_{\mathcal{Q}} \cup \mathcal{C}_{\mathcal{R}}$, where $\mathcal{C}_{\mathcal{Q}}$ represents the constraints related to environments, such as "the scroll cannot be lifted", and $\mathcal{C}_{\mathcal{R}}$ represents the constraints stemming from the robot's physical limitations, encompassing aspects like the robot's workspace boundary and the extent of its skills. Let the robot embodiment space be $\mathcal{R}$. Each constraint $C \in \mathcal{C}$ can be activated based on different combination of $\mathcal{Q}$ and $\mathcal{R}$. The API $F_{\mathcal{C}}$ helps parse the constraints $\mathcal{C}$ into a constraint description $L_{\mathcal{C}} = F_{\mathcal{C}}(\mathcal{C})$. The user will provide the task $L_{\mathcal{T}}$ in natural language. The concatenated language description $L = \{L_{\mathcal{T}}, L_{\mathcal{Q}}, L_{\mathcal{C}}\}$ serves as the query input to our method.

**Hierarchical Policies for Robot Tool Use.** We consider a Markov Decision Process $\mathcal{M}$ defined by a tuple $(\mathcal{S}, \mathcal{A}, p, r, \rho_0)$, representing the state space, action space, transition dynamics, reward function, and initial state distribution, respectively. We use a two-level hierarchy consisting of a set of parameterized skills and a high-level controller. For each robot embodiment $R \in \mathcal{R}$, we assume that there is a set of parameterized skills $\Pi^R = \{\pi_i^R\}_{i=1}^N$. Each skill receives a parameter $x \in \mathcal{X}_i$, where $\mathcal{X}_i$ is the parameter space of skill $i$. The skill $\pi_i^R(x)$ generates a squence of low-level actions $(a_1, \cdots, a_t, \cdots), a_t \in \mathcal{A}$. A parameterized skill can be moving the robotic arm's end effector to a targeted position $x$. The high-level controller outputs $(H, X)$, where $H = (h_1, \ldots, h_k, \ldots)$ is the skill sequence, $h_k \in [N]$ is the skill at plan step $k$, $X = (x_{h_1}^{(1)}, \ldots, x_{h_k}^{(k)}, \ldots)$ is the corresponding parameter sequence, and $x_{h_k}^{(k)} \in \mathcal{X}_{h_k}$ denotes the parameter for skill $h_k$. Given a language description $L$ for the tool-use tasks, our goal is to generate a code $\tau((H, X), \Pi, L)$ that can solve the task by calling a sequence of parameterized skills meanwhile providing their parameters. Considering the feasible solution $\tau$ may not be unique and potentially involve manipulating different numbers of objects as tools, besides task completion, we also desire a simple plan that interacts with a minimal number of objects in the scene.

## 3.2 ROBOTOOL: CREATIVE ROBOT TOOL USE WITH LARGE LANGUAGE MODELS

We propose RoboTool, that solves hybrid discrete-continuous planning problems with constraints through creative tool use in a zero-shot manner. RoboTool takes *natural language instructions* as

inputs, making it user-friendly and outputs *executable codes* calling the robot's parameterized skills. RoboTool maintains a hierarchical structure consisting of four key components (Fig. 2), including an *Analyzer*, a *Planner*, a *Calculator* and a *Coder*, each is a LLM handling one functionality.

### 3.2.1 ANALYZER

Humans can clearly identify crucial concepts that will affect the task plan (Weng et al., 2023). For instance, when placing a book on a bookshelf, we use the book's dimensions, available shelf space, and slot height to determine whether the task is feasible. *Can we endow robots with such reasoning capability to identify key concepts before detailed planning?* We seek to answer the question by utilizing LLMs, which store a wealth of knowledge about objects' physical and geometric properties and human common sense. We propose the *Analyzer*, powered by LLMs, which extract the key concepts and their values that are crucial to determine task feasibility. *Analyzer* is fed with a prompt that structures its response in two segments: an analysis section elucidating its thinking process and a description section listing the key concepts alongside their values and the related constraint. An example output of *Analyzer* is shown in Fig. 3. We add the content in the description section of the *Analyzer* output to the original descriptions $L$ to construct the key concept augmented description $L^*$ for downstream modules.

```
<start of analysis>
The key feature that affects the
feasibility of the plan is the gap
between sofa_1 and sofa_2...

To calculate the gap, we need to
consider the x-axis positions of the
two sofas and their sizes. The center
of sofa_1 is at x=0.0 and its size
along the x-axis is 1.5m, so its edge
is at x=0.0+1.5/2=0.75m...

Therefore, the gap between the two
sofas is 1.15m - 0.75m = 0.4m, which
is larger than the maximum gap the
robot can walk across (0.1m).
<end of analysis>

<start of description>
The key feature is the gap between
sofa_1 and sofa_2 which is 0.4m,
since the robot can only walk across
a gap smaller than 0.1m. According to
the initial configuration, the
constraint is violated initially.
<end of description>
```

Figure 3: Analyzer output.

It's worth noting that the LLMs' internalized prior knowledge autonomously determines the selection of these key concepts, and there is no prerequisite to delineating a predefined set of concepts. Moreover, *Analyzer* can extract explicit concepts provided in the description in $L$, such as the objects' positions and related workspace ranges, and implicit concepts that require calculations based on provided numerical information, such as the gap width between two objects as in Fig. 3.

### 3.2.2 PLANNER

Motivated by the strong task decomposition capability of LLMs (Ahn et al., 2022; Huang et al., 2022a), we propose to use an LLM as a *Planner* to generate a plan skeleton $H$ based on the key concept augmented language description $L^*$. The response of the *Planner* contains a description section and a plan section, as shown in Fig. 4. *Planner* first describes each object's properties and possible roles in finishing the task, showing the reasoning process and constructing an abstract plan in the description section. *Planner* then generates a detailed plan skeleton based on the parameterized skills in the plan section. We provide *Planner* with a prompt describing each parameterized skill, formats of the response, and rules for the two sections.

```
<start of description>
[SOFA_1]: ...
[SOFA_2]: ...
[SURFBOARD]: The surfboard is light enough for the robot to
push and can be used as a bridge to cross the gap between the
sofas.
[STRIP_OF_CLOTH]: The strip of cloth is on sofa_1 but it is too
thin and small to be useful in this task.
[Key Feature and constraints]: The key feature is the gap
between sofa_1 and sofa_2...
[Abstract Plan]: The robot should first push the surfboard to
the edge of sofa_1...
<end of description>

<start of plan>
- Use the 'get_position' skill to...
- Use the 'push_to_position' skill to push the surfboard...
...
<end of plan>
```

Figure 4: Planner output.

We observe that *Planner* can automatically generate a plan by utilizing objects within the environment as intermediate tools to complete a task, with the help of the key concept augmentation. Examples include "using a box as a stepping stone" or "using a hammer to drag the milk carton in the workspace." *Planner* can discover functionalities of the objects beyond their standard affordances, demonstrating the *creative tool-use capability* by reasoning over the objects' physical and geometric properties. In addition, *Planner* can generate a long-horizon plan, especially in handling tasks requiring multiple tools sequentially. For instance, it can generate a plan consisting of 15 plan steps for the "Can-Grasping" task as in Fig. 5.

### 3.2.3 CALCULATOR

Existing literature shows that LLMs' performance tends to decline when they operate across varied levels of abstractions (Liang et al., 2023; Yu et al., 2023). Inspired by these findings, we introduce a *Calculator*, a standalone LLM for calculating the desired parameters for the parameterized low-

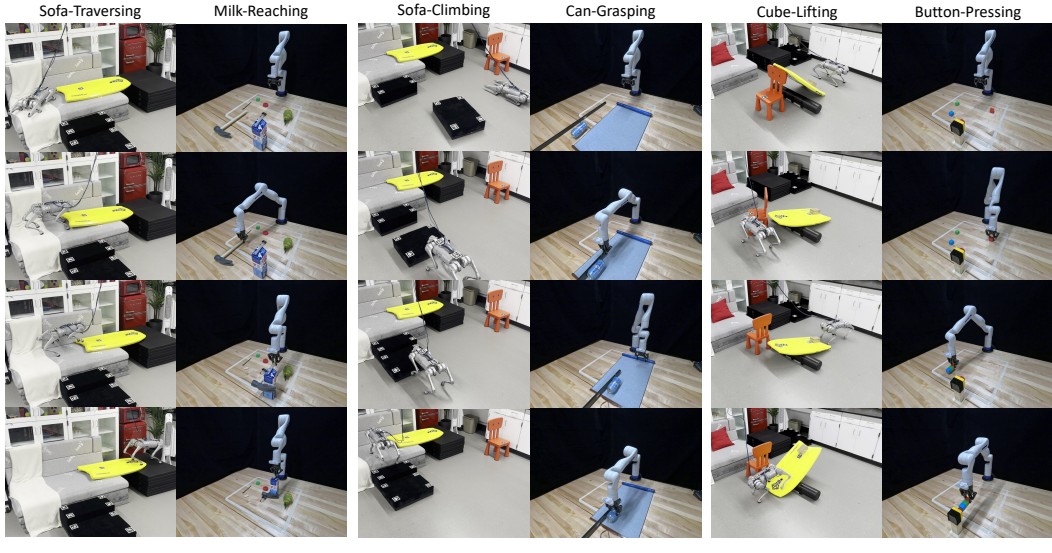

|  |  |  |
|---|---|---|
| (a) Tool selection | (b) Sequential tool use | (c) Tool manufacturing |

Figure 5: Visualization of RoboTool's creative tool-use behaviors. (a) Tool selection. The quadrupedal robot needs to select the surfboard over a strip of cloth and push it to bridge the gap (**Sofa-Traversing**). The robotic arm needs to choose the hammer among many options and use it as a hook to pull the milk carton into the workspace (**Milk-Reaching**). (b) Sequential tool use. The quadrupedal robot needs to push a small box against a large box adjacent to the sofa and use the small box as the initial stepstone and the large box as the intermediate stepstone to climb onto the sofa (**Sofa-Climbing**). The robotic arm needs to pick up a stick, push a can onto a strip of paper, and then pull the paper closer with the can on it (**Can-Grasping**). (c) Tool manufacturing. The quadrupedal needs to identify the hidden lever structure in the environment and push away a chair supporting one end of the lever so that it can activate the lever arm and lift a heavy cube (**Cube-Lifting**). The robotic arm needs to assemble magnetic blocks to create a stick to press a button outside its workspace (**Button-Pressing**).

level skill at each plan step, denoted as $X$ in Sec. 3.1. *Calculator* processes both the key concept augmented description $L^*$ and the *Planner*-generated plan skeleton $H$ for its calculation. Similar to the *Analyzer* and *Planner*, it generates a response with two sections, including a description section showing its calculation process and an answer section containing the numerical values. These numerical outcomes, representing target positions, are then integrated into each corresponding step of the plan skeleton. We provide *Calculator* with multiple exemplars and rules to help deduce target positions. *Calculator* can generate navigation target positions for quadrupedal robots and push offsets for robotic arms to manipulate objects.

### 3.2.4  CODER

Finally, we introduce a *Coder* module that transforms the plan $(H, X)$ into an executable code script $\tau$ that invokes robot low-level skills, perception APIs, and built-in Python libraries to interact with the environment. We provide *Coder* with the definitions of each low-level skill and template-related rules. Although *Coder* generates the script in an open-loop manner, the produced code is inherently designed to receive feedback from the environment. It achieves this through built-in skills like "`get_position`", granting a certain level of responsiveness to environmental changes.

## 4  CREATIVE ROBOT TOOL USE BENCHMARK

Unlike conventional tool use, creative tool use, also termed as flexible tool use, is recognized by some as an indication of advanced intelligence, denoting animals' explicit reasoning about tool applications contingent on context (Call, 2013). From a general problem-solving perspective, Fitzgerald et al. (2021) further characterized human creativity attributes, including improvisation in the absence of typical tools, use of tools in novel ways, and design of innovative tools tailored for new tasks. In this work, we aim to explore three challenging categories of *creative tool use* for robots: **tool selection**, **sequential tool use**, and **tool manufacturing** (Qin et al., 2023), and design six tasks for two different robot embodiments: a quadrupedal robot and a robotic arm. The details of each task are as shown in Fig. 5 and Sec. E, and the constraints of each task are listed in Tab. 2.

Table 1: Success rates of RoboTool and baselines. Each value is averaged across 10 runs. All methods except for **RoboTool (Real World)** are evaluated in simulation.

| | Milk-Reaching | Can-Grasping | Button-Pressing | Sofa-Traversing | Sofa-Climbing | Cube-Lifting | Average |
|---|---|---|---|---|---|---|---|
| **RoboTool** | **0.9** | **0.7** | **0.8** | **1.0** | **1.0** | **0.6** | **0.83** |
| **RoboTool w/o Analyzer** | 0.0 | 0.4 | 0.2 | **1.0** | 0.7 | 0.2 | 0.42 |
| **RoboTool w/o Calculator** | 0.0 | 0.1 | **0.8** | 0.3 | 0.0 | 0.3 | 0.25 |
| **Planner-Coder** | 0.0 | 0.2 | 0.5 | 0.1 | 0.0 | 0.4 | 0.20 |
| **Coder** | 0.0 | 0.0 | 0.0 | 0.0 | 0.0 | 0.4 | 0.07 |
| **RoboTool (Real World)** | 0.7 | 0.7 | 0.8 | 0.7 | 0.8 | 0.9 | 0.77 |

Table 2: RoboTool's proposed key concept accuracy in simulation. Each value is averaged across 10 runs.

| | Key Concept and Violated Constraints | Accuracy |
|---|---|---|
| **Milk-Reaching** | Milk's position is out of robot workspace. | 1.0 |
| **Can-Grasping** | Can's position is out of robot workspace. | 1.0 |
| **Button-Pressing** | Button's position is out of robot workspace. | 0.9 |
| **Sofa-Traversing** | Gap's width is out of robot's walking capability. | 1.0 |
| **Sofa-Climbing** | Sofa's height is out of robot's climbing capability. | 0.8 |
| **Cube-Lifting** | Cube's weight is out of robot's pushing capability. | 1.0 |

- **Tool selection** (Sofa-Traversing and Milk-Reaching) requires the reasoning capability to choose the most appropriate tools among multiple options. It demands a broad understanding of object attributes such as size, material, and shape, as well as the ability to analyze the relationship between these properties and the intended objective.

- **Sequential tool use** (Sofa-Climbing and Can-Grasping) entails utilizing a series of tools in a specific order to reach a desired goal. Its complexity arises from the need for long-horizon planning to determine the best sequence for tool use, with successful completion depending on the accuracy of each step in the plan.

- **Tool manufacturing** (Cube-Lifting and Button-Pressing) involves accomplishing tasks by crafting tools from available materials or adapting existing ones. This procedure requires the robot to discern implicit connections among objects and assemble components through manipulation.

## 5 EXPERIMENT RESULTS

We aim to investigate whether RoboTool possesses various types of creative tool-use capabilities by evaluating it on the benchmark outlined in Sec. 4. We build both simulation and real-world platforms detailed in Sec. 5.1 and compare RoboTool with four baselines described in Sec. 5.2. We measure the task success rates to understand the performance of RoboTool in Sec. 5.3 and analyze the effect of RoboTool's modules through error breakdown in Sec. 5.4. We then dive deeper into the role of *Analyzer* and show that it enables discriminative creative tool-use behaviors in Sec 5.5.

### 5.1 EXPERIMENT SETUP

**Robotic Arm.** We test RoboTool with a Kinova Gen3 robotic arm (details in Sec. C). In simulation, we build tasks based on robosuite (Zhu et al., 2020) and assume known object positions and sizes. In real-world experiments, we employ OWL-ViT (Minderer et al., 2022) to obtain 2D locations and bounding boxes for each object. In both platforms, the robot maintains a skill set as ["get_position", "get_size", "open_gripper", "close_gripper", "move_to_position", "get_workspace_range"]. We use skills without explicitly listing the object-centric movements caused by the "move_to_position" skill, such as pushing or picking.

**Quadrupedal Robot.** We test RoboTool with a Unitree Go1 quadrupedal robot (details in Sec. B). The simulation experiments for quadrupedal robots are evaluated based on the generated code and through human evaluations. In real-world experiments, considering the relatively large workspace compared with the tabletop setting with the robotic arm, we use AprilTags (Olson, 2011) affixed to each object in real-world experiments to get the object's positions. Each skill in real-world experiments is equipped with skill-specific motion planners to generate smooth and collision-free velocity commands for different walking modes of Go1. For both simulation and real-world experiments, the quadrupedal robot's skill set is ["get_position", "get_size", "walk_to_position", "climb_to_position", "push_to_position"].

## 5.2 BASELINES

We compare RoboTool with four baselines, including one variant of Code-as-Policies (Liang et al., 2023) and three variants of our proposed RoboTool.

- **Coder.** It takes the natural language instruction as input and directly outputs executable code. It is a variant motivated by Code-as-Policies (Liang et al., 2023). This baseline demonstrates the combinatorial effect of the other three modules in RoboTool.
- **Planner-Coder.** It removes the *Analyzer* and the *Calculator* in RoboTool. This baseline demonstrates the combinatorial effect of the *Analyzer* and the *Calculator* modules.
- **RoboTool without Analyzer.** The *Planner* directly takes the language instruction as input. This baseline reveals the effect of the *Analyzer* in the downstream planning.
- **RoboTool without Calculator.** The *Coder* directly takes the response of the *Planner*. This baseline demonstrates the effect of the *Calculator* module.

We evaluate RoboTool both in simulation and in the real world while only evaluating of baselines in simulation given their relatively low success rates in simulation. RoboTool's prompts are in Sec. D.

## 5.3 CAN ROBOTOOL ACHIEVE CREATIVE TOOL USE?

We present the quantitative success rates of RoboTool and baselines in Tab. 1 and real-world qualitative visualizations of RoboTool in Fig. 5. RoboTool consistently achieves success rates that are either comparable to or exceed those of the baselines across six tasks in simulation. RoboTool's performance in the real world drops by 0.1 in comparison to the simulation result, mainly due to the perception errors and execution errors associated with parameterized skills, such as the quadrupedal robot falling down the soft sofa. Nonetheless, RoboTool (Real World) still surpasses the simulated performance of all baselines. Considering that the tasks in Sec. 4 are infeasible without manipulating objects as tools, we show that RoboTool can successfully enable robot tool-use behaviors. Moreover, as visualized in Fig. 5, RoboTool transcends the standard functionalities of objects and creatively capitalizes on their physical and geometric properties, including materials, shapes, and sizes. Here are some highlights of the creative tool-use behaviors.

**Piror Knowledge.** In the Milk-Reaching task (Fig. 5a), RoboTool leverages LLM's prior knowledge about all the available objects' shapes and thus improvisationally uses the hammer as an L-shape handle to pull the milk carton into the workspace.

**Long-horizon Planning.** In the Can-Grasping task (Fig. 5b), RoboTool sequentially uses the stick to push the can onto the scroll and then drag the scroll into the workspace with the can on it. This reveals RoboTool's long-horizon planning capability by generating a plan with as many as 15 steps.

**Hidden Mechanism Identification.** In the Cube-Lifting task with the quadrupedal robot (Fig. 5c), RoboTool identifies the potential existence of a mechanism consisting of the yoga roller as the fulcrum and the surfboard as the lever. RoboTool first constructs the lever by pushing the chair away, then activates the lever by walking to one end of the lever, and finally lifts the cube. It illustrates that RoboTool can not only fabricate a tool from available objects but also utilize the newly created tool.

RoboTool without Analyzer performs worse than RoboTool while better than RoboTool without Calculator. Moreover, they perform better than baselines lacking *Analyzer* and *Calculator*, including Planner-Coder and Coder. These observations show that both *Analyzer* and *Calculator* are critical in achieving high success rates, and *Calculator* plays a more important role in tasks that require accurate positional offsets such as Milk-Reaching, Can-Grasping and Sofa-Climbing.

## 5.4 ERROR BREAKDOWN

We further analyze what causes the failure of RoboTool and baselines based on simulation experiments. We define three types of errors: tool-use error, logical error, and numerical error. The tool-use error indicates whether the correct tool is used. The logical error mainly focuses on the planning error, such as using tools in the wrong order or ignoring the constraints provided. The numerical error includes calculating the wrong target positions or adding incorrect offsets. We show the error breakdown averaged across six tasks in Fig. 6a. The results show that the *Analyzer* helps reduce the tool-use error when comparing RoboTool and RoboTool without Analyzer. *Calculator* significantly reduces the numerical error when comparing RoboTool, RoboTool without Calculator and Planner-Coder. We provide per-task error breakdown results in Sec. A.

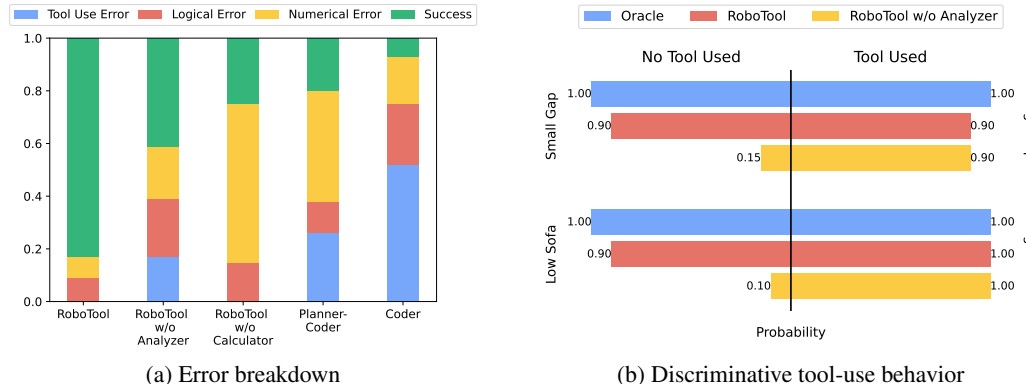

(a) Error breakdown   (b) Discriminative tool-use behavior

Figure 6: (a) Error breakdown of RoboTool and baselines. (b) Discriminative tool-use behavior is enabled by *Analyzer*'s explicit reasoning about the key concepts.

## 5.5 How does Analyzer Affect the Tool-Use Capability?

**Key Concept Identification Accuracy.** We show the accuracy of the proposed key concept in Tab. 2, based on whether the *Analyser* correctly returns the key concept, the value of the key concept, and the related constraint. The target responses are provided by human. The results show that *Analyzer* could correctly identify the key concept that affects the plan's feasibility and accurately calculate the key concepts' value. For instance, in the Sofa-Traversing task, the key concept is identified as the distance between the boundaries of the two sofas, which is the gap width the robot needs to cover. Moreover, the *Analyzer* could link the key concept with the robot's limit: the quadrupedal robot can only walk across a gap over 0.1m.

**Discriminative Tool-use Capability.** Given the impressive creative tool-use capability of RoboTool, we want to investigate further whether RoboTool possesses the discriminative tool-use capability, which is using tools when necessary and ignoring tools when the robot can directly finish tasks without the need to manipulate other objects. We choose Sofa-Traversing and Sofa-Climbing to test the discriminative tool-use capability. For Sofa-Traversing, we compare the rate of tool use in scenarios with large gaps where using tools is necessary, against scenarios with small gaps that allow the robot to traverse to another sofa without using tools. For Sofa-Climbing, we evaluate the tool-use rate in scenarios where a high-profile sofa requires the use of boxes as stepstones, in contrast to low-profile sofas, in which the robot can ascend directly without assistance.

We compare the RoboTool with an Oracle that can derive the most efficient plan and the baseline RoboTool without Analyzer, and present the main results in Fig. 6. In both sets of tasks, RoboTool tends not to use tools when unnecessary (Small Gap and Low Sofa), demonstrating more adaptive behaviors given different environment layouts. In contrast, without the help of *Analyer*, the baseline tends to use tools in all four scenarios, dominated by the prior knowledge in LLMs. These observations show that *Analyser* helps enable the discriminative tool-use behavior of RoboTool.

## 6 Conclusion

We introduce RoboTool, a creative robot tool user powered by LLMs that enables solving long-horizon planning problems with implicit physical constraints. RoboTool contains four components: (i) an "Analyzer" that discerns crucial task feasibility-related concepts, (ii) a "Planner" that generates creative tool-use plans, (iii) a "Calculator" that computes skills' parameters, and (iv) a "Coder" that generates executable code. We propose a benchmark to evaluate three creative tool-use behaviors, including tool selection, sequential tool use, and tool manufacturing. Through evaluating on the creative tool use benchmark, we show that RoboTool can identify the correct tool, generate precise tool-usage plans, and create novel tools to accomplish the task. We compare our method to four baseline methods and demonstrate that RoboTool achieves superior performance when the desired tasks require precise and creative tool use.

**Limitations.** Since we focus on the tool-use capability of LLMs at the task level in this paper, we rely on existing APIs to process visual information, such as describing the graspable points of each object and summarizing the scene. It is possible to integrate vision language models to replace the designed API to get the affordance for each object similar to VoxPoser (Huang et al., 2023b). In addition, we highlight that the proposed method serves as a planner, specializing in executable plan creation, not an execution framework. Reactive execution with a feedback loop could be achieved by integrating hybrid shooting and greedy search into our method, such as in Lin et al. (2023b).

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

# A ADDITIONAL EXPERIMENT RESULTS

We provide additional error breakdown results in Fig. 7. We observe that different modules play different roles in various tasks. For instance, in the Milk-Reaching task, the Planner-Coder baseline is dominated by the tool use error without knowing using the hammer as the tool to drag the milk into the workspace. In this case, *Analyzer* helps reduce the tool use error significantly. In contrast, in the Cube-Lifting task, most of the generated plans could construct a lever by pushing away the chair. However, the baselines tend to ignore the weight of the cube and assume that the dropping surfboard could automatically lift the cube. In this case, the *Analyzer* helps reduce the logical error. While in other tasks, the *Calculator* becomes quite important, especially in Can-Grasping, Sofa-Climbing, and Milk-Reaching.

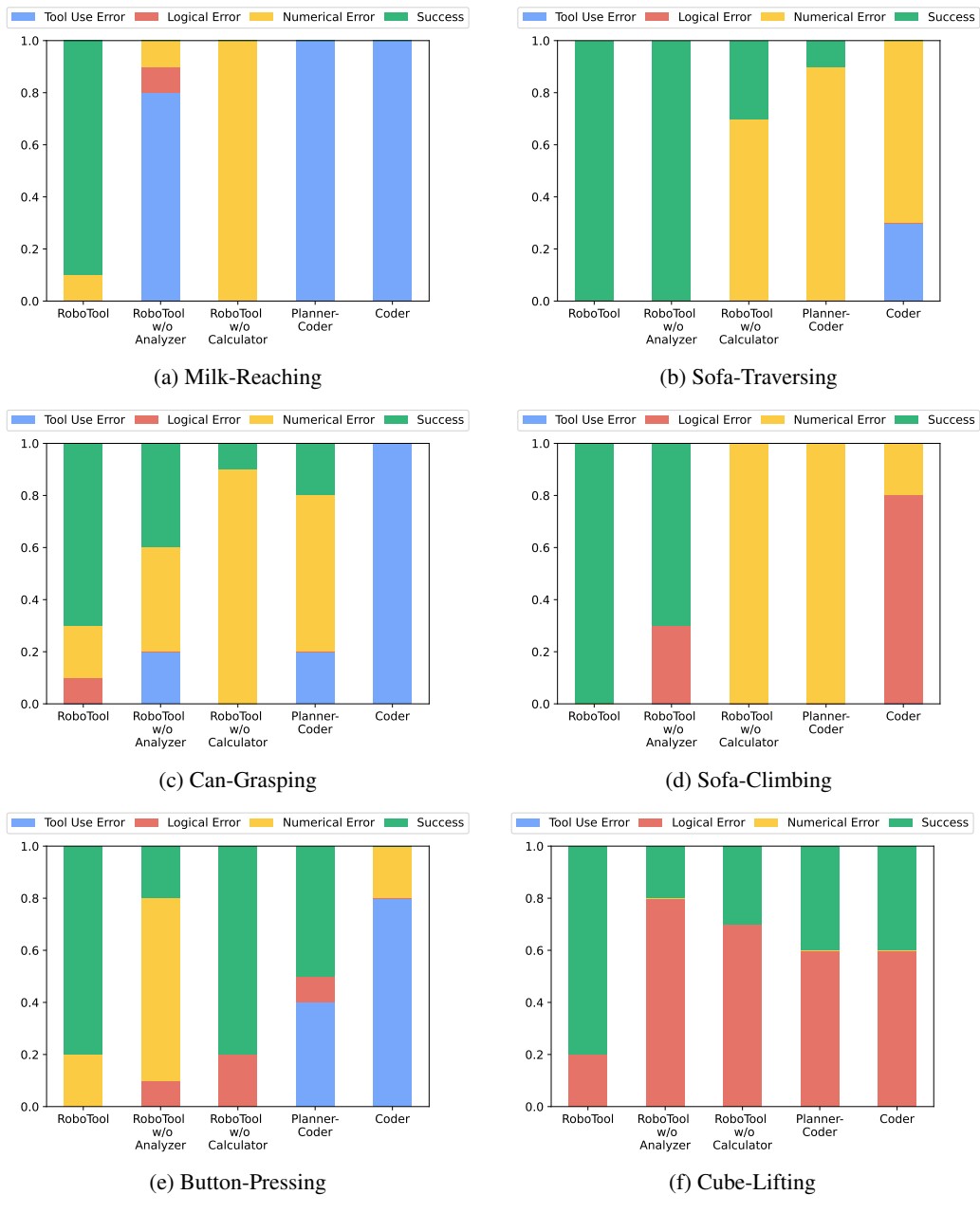

Figure 7: Error breakdown for each creative tool-use task.

## B    REAL-WORLD SETUP FOR QUADRUPEDAL ROBOT

In the quadrupedal robot environment setup, several objects with which the robot can interact are presented in Fig. 8. These include two blocks of varying heights, two sofas positioned adjacently with gaps between them, a chair, a surfboard, and a yoga roller. Additionally, two ZED2 cameras are situated at the top-left and top-right of the environment to capture the positions of the robot and other objects. These April tags are identifiable by the two ZED 2 cameras situated at the top-left and top-right of the environment, enabling the computation of the object's position using the PnP algorithm Fischler & Bolles (1981).

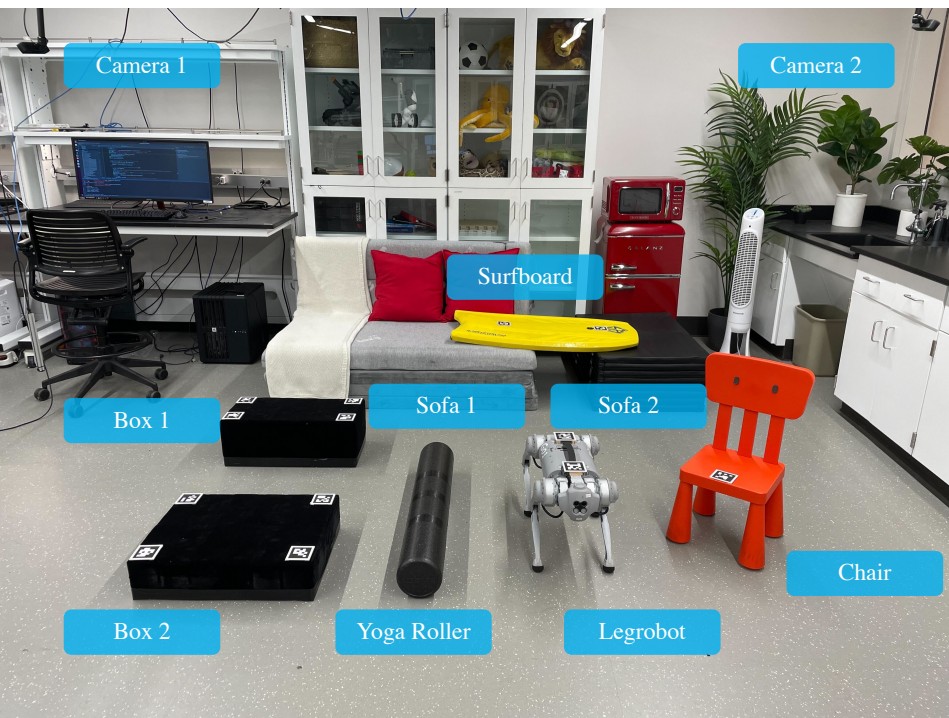

Figure 8: The figures illustrate the quadrupedal robot environment setup including object names and geometries. The image shows various objects with which the robot can interact. The above-described objects are labeled with names in the figure.

The quadrupedal robot possesses five skills within its skill set: `move_to_position`, `push_to_position`, `climb_to_position`, `get_position`, `get_size`.

### B.1    MOVE_TO_POSITION

Upon invoking this skill, the quadrupedal robot navigates to the target position from its current location, avoiding obstacles present in the scene. The movement is facilitated using the built-in trot gait in continuous walking mode from Unitree. Trajectories are generated using the informed RRT* method Gammell et al. (2014) to prevent potential collisions during trajectory planning. Fig. 9 illustrates an example of a trajectory produced by the motion planner and demonstrates the robot's movement along this path.

### B.2    PUSH_TO_POSITION

When this skill is called, the quadrupedal robot pushes an object to the target location following this sequence, also as demonstrated in Fig. 10:

1. **Rotate Object:** The quadrupedal robot initially attempts to rotate the object until its rotation along the z-axis aligns with the target.

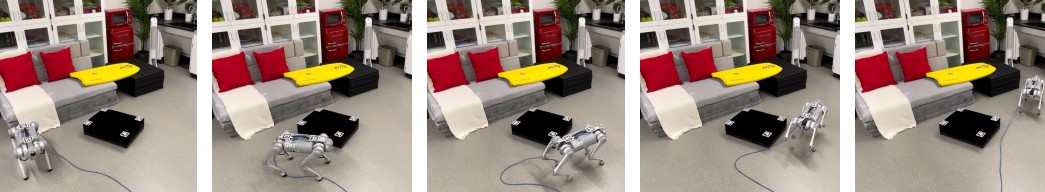

Figure 9: This figure illustrates the robot's moving skill, the robot reaches the target location while avoiding an obstacle on the path. The collision-free trajectory is generated by an informed RRT* path planner.

2. **Push along y-axis:** The quadrupedal robot subsequently attempts to push the object along the y-axis until the object's y-position matches the target.

3. **Push along x-axis:** Finally, the quadrupedal robot pushes the object along the x-axis until the object's x-position meets the target.

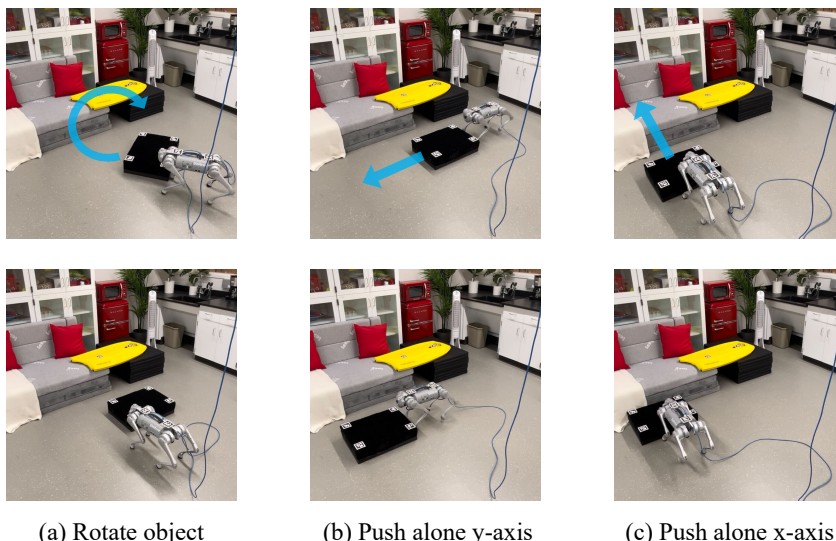

(a) Rotate object       (b) Push alone y-axis       (c) Push alone x-axis

Figure 10: This figure demonstrates the robot's object-pushing skill. (a) First, the robot rotates the object by pushing one corner. (b) Then, it pushes the object along the y-axis, (c) followed by the x-axis, until the object reaches its designated location.

### B.3 CLIMB_TO_POSITION

This skill enables the robot to climb to the desired location utilizing the built-in stair-climbing mode from Unitree. Path planning is disabled when this skill is invoked because the robot is able to move above obstacles.

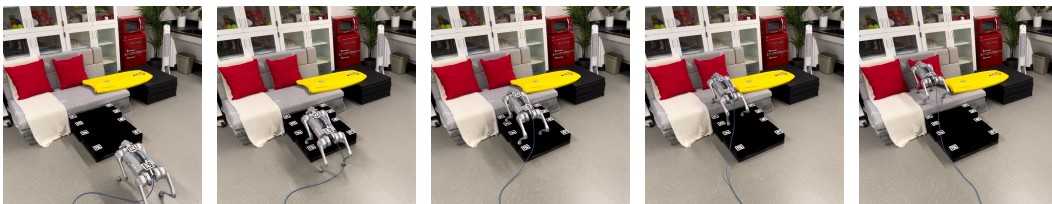

Figure 11: An illustration showcasing the robot's climbing skill, the robot successfully ascends onto a sofa by navigating through box 1 and box 2 of differing heights.

Fig. 11 illustrates an instance where the robot climbed onto a sofa by climbing on two boxes of varying heights.

### B.4  GET_POSITION

The position of each object is estimated using AprilTags affixed to them. These AprilTags are identifiable by the two ZED 2 cameras, enabling the computation of the object's position using the PnP algorithm (Fischler & Bolles, 1981). Fig. 12 shows the estimated positions of some objects from one camera.

### B.5  GET_SIZE

The bounding boxes of the objects are pre-measured and stored in a database. Each time this function is invoked, it returns the object's size based on its position and orientation.

Fig. 12 illustrated some object bounding boxes estimated from one camera.

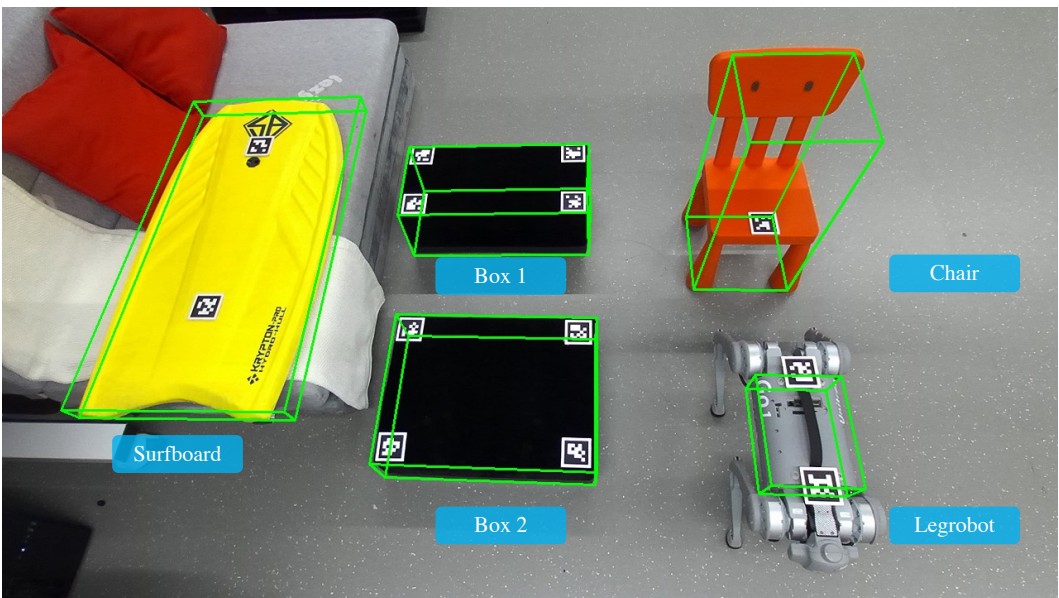

Figure 12: This image illustrates the estimated positions and bounding boxes of various objects, as computed by the PnP algorithm by capturing AprilTags placed on each object. The objects shown include box 1, box 2, a surfboard, a quadrupedal robot, and a chair.

## C  REAL-WORLD SETUP FOR THE ROBOTIC ARM

We test RoboTool using a Kinova Gen3 Robot arm with 7 degrees of freedom and a two-fingered gripper. In real-world experiments, we applied the OWL-ViT Minderer et al. (2022) to obtain 2D locations and bounding boxes for each object. We did this by capturing a slightly tilted top-down view of the scene. Next, we converted the coordinates of the bounded image from 2D to both world coordinates and robot coordinates. Finally, we combined the depth information of each detected object with the transformed 2D bounding box in robot coordinates to calculate the complete 3D position and size of the objects in the scene.

We assume the graspable point of each object is given to RoboTool. Graspable point of objects is a long-standing and active research field in robotics (Fang et al., 2020; Lin & Sun, 2015; Myers et al., 2015; Song et al., 2010; Ek et al., 2010; Song et al., 2015). In this work, we focus on the high-level planning capability of LLMs rather than the low-level grasping policy.

In the robot arm environment setup, the tasks focus on table-top manipulations. such as Button-Pressing, Milk-Reaching, and Can-Grasping. Tasks are executed by the combination of the

following skills: `move_to_position`, `open_gripper`, `close_gripper`, `get_position`, `get_size`

## C.1 MOVE_TO_POSITION

Upon invoking the `move_to_position` skill, the built-in Kinova high-level planner would generate waypoints along the Euclidean distance direction between the current tool pose and target position. However, there are some constraints introduced by certain object scenes. The detailed motion planning paths are shown in Fig. 13 and described as follows:

1. **Milk-Reaching:** Due to the geometric features of object *hammer*, which its centre does not represent grasping point of the object, we added an object-specfic offset in both x and y axes to the motion planner when grasping the hammer. All the other motion bahavior are generated by RoboTool and directly executed by Kinova high-level motion planner.

2. **Can-Grasping:** Under the object settings, we have pre-scripted a collsion-free path given the target position. Instead of moving along Euclindean distance vector, we assume the scene is in grid world settings where the arm agent could only move in one direction once. The motion of approching target objects is start with Y, followed by X, and then Z.

3. **Button-Pressing:** For the magnetic toy cubes geometries, only the flat surface can be attached firmly. To resolve the instability, we assume the Button-Pressing scene is in grid world settings where the arm agent could only move in one direction once. The motion of approching target objects is start with Z, followed by Y, and then X.

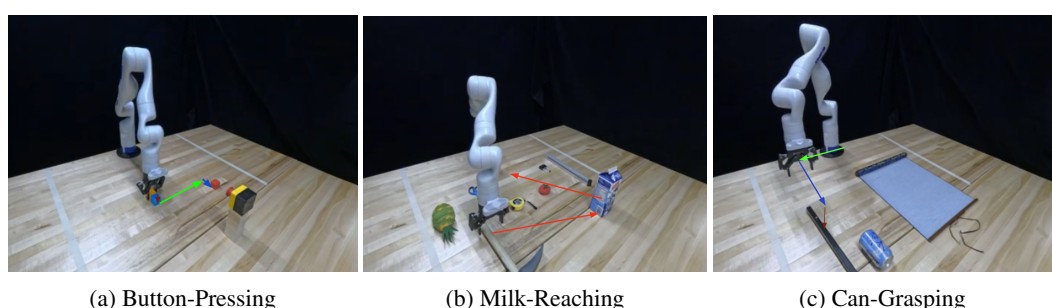

(a) Button-Pressing          (b) Milk-Reaching          (c) Can-Grasping

Figure 13: The figure demonstrates the scripted execution order for an arm motion planner. The red arrows in (b) show the euclidean distance vector motion. (a) and (c) shows the scripted moving orders on each axis.

## C.2 GET_POSITION

When invoking the `get_position` function, we employed the OWL-ViT methodology as detailed in the reference Minderer et al. (2022). This approach allowed us to derive 2D bounding boxes encompassing the objects within the scene. This was achieved by capturing a slightly slanted top-down perspective of the environment. Following this, we conducted a conversion of the bounded image's coordinates from 2D to both world coordinates and robot coordinates. Subsequently, we fused the depth information from stereo input to each identified object with the transformed 2D bounding box represented in robot coordinates. As a result, we were able to calculate the comprehensive 3D positions of the objects within the scene. add detection picture. Fig. 14 presents an example of various object positions as detected by the OWL-ViT detector.

## C.3 GET_SIZE

While this function is invoked, the size of objects can also be captured using methods as described in function `get_position`. The output is in three dimensions, which include the width, length, and height of the objects. Fig. 14 also presents the bounding box of various object positions.

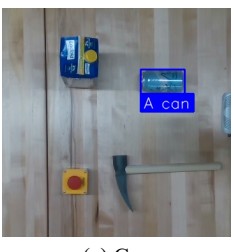 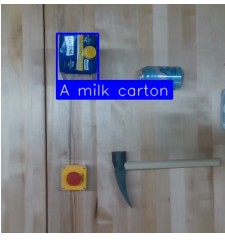 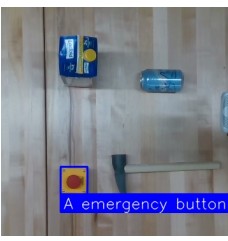 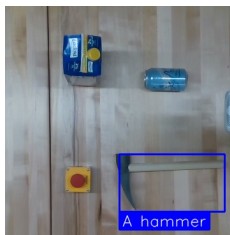

|        (a) Can        |     (b) Milk Carton     |    (c) Emergency button    |      (d) Hammer      |

Figure 14: This image demonstrates the detection capabilities of the OWL-ViT detector. The detector successfully identifies various objects along with their respective positions and bounding boxes on the table.

### C.4  OPEN & CLOSE GRIPPER

This would connect with Kinova API on closing and opening the gripper. While closing the gripper, the gripper finger distance does not need explicit scripts, where the Kinova built-in gripper sensors would automatically grasp the object under pre-set pressure.

## D  PROMPTS

### D.1  ROBOTIC ARMS

#### D.1.1  ANALYZER PROMPT FOR ROBOT ARM EXPERIMENTS

> Answer template:
> <start of analysis>
> <end of analysis>
> <start of description>
> The key feature is the weight of the box which is 10kg, since the robot can only push a box with a maximum weight of 5kg.
> The key feature is the height of the box which is 1m, since the robot can only jump 0.5m.
> <end of desctiption>
> Remember:
> - Think step by step and show the steps between <start of analysis> and <end of analysis>.
> - Return the key feature and its value between <start of description> and <end of description>.
> - The key features may be the 3D spatial information that may not be directly included in the [Scene Description] but needs further calculation.
> If you understand, just say Yes. Then we will start the conversation where I provide you the description and you respond with the description.

#### D.1.2  PLANNER PROMPT FOR ROBOT ARM EXPERIMENTS

> You must follow the following answer template:
> Given a object list [OBJECT1, OBJECT2, OBJECT3]
> <start of high-level idea>
> [OBJECT1]: ...
> [OBJECT2]: ...
> [OBJECT3]: ...
> The spatial relationship between [OBJECT1] and [OBJECT2]: ...
> The spatial relationship between [OBJECT2] and [OBJECT3]: ...
> The spatial relationship between [OBJECT1] and [OBJECT3]: ...
> The abstract plan: ...
> <end of high-level idea>
> <start of detailed plan>
> <end of detailed plan>

Common rules for high-level idea:
- You must only choose [OBJECT] from the object list.
- You must show the physical properties of objects, their affordances, and their roles in completing the task.
- You must reason about the relative positions of the objects along each axis. You must describe the spatial relationship between each pair of objects in the object list first based on numerical positions along each x, y and z axis. For example, whether the objects are in contact, and whether one object is on top of another object etc.
- You must reason based on each object's properties and develop an abstract plan with the highest confidence.
- Think about how many objects to use to finish the task.
- You must strictly adhere to the constraints provided in the scene information.
- You must think step by step and show the thinking process. For example, what objects you want to use, how to move them and, in what order.
- You need to ensure that the objects are in the right positions for you plan.
You are a robot arm with a gripper as the end effector. The gripper is open initially. You have a skill set containing the following skills:
* 'move_to_position': Move the gripper to a position. It uses the goal-conditioned policy. You can use it to move to a target position. Moreover, you can use it with proper tools for manipulation tasks. You cannot rotate the gripper. You can only translate the gripper.
* 'open_gripper': open the gripper before grasping an object.
* 'close_gripper': close the gripper to grasp an object.
* 'get_position': get the position of an object.
Common rules for detailed plan:
- You must plan according to the high-level idea.
- You must use existing skills.
- You must make each plan step only call one skill at once and be as atomic as possible.
- Before planning every step, rehearse the relevant [OBJECT]'s [REHEARSED_POSITION] after the step.
- You must not knock down any objects.
- You must get the updated [OBJECT]'s position again if [OBJECT] has moved since the last 'get_position([OBJECT])'.
Individual rules with example answers for detailed plan:
Rule: To use a grasped [OBJECT1] to move another [OBJECT2]: in the first step, you must make sure [OBJECT1]'s bounding box is adjacent to the [OBJECT2]'s bounding box to ensure that they are in contact. [OBJECT2]'s position does not change because of the contact. In the next step, you should move the grasped [OBJECT1] to push [OBJECT2].
Example:
- Use the 'move_to_position' to move the grasped [OBJECT1] to make contact with [OBJECT2]. The [OBJECT2]'s rehearsed position is [REHEARSED_POSITION] after this step.
- Use the 'move_to_position' to push the [OBJECT2] ...
Rule: To push an [OBJECT] into the workspace, the xy position of [OBJECT]'s [REHEARSED_POSITION] must be as close to [0.0, 0.0] as possible.
Example:
- Use the 'move_to_position' to push the [OBJECT] into the workspace. The [OBJECT]'s rehearsed position is [REHEARSED_POSITION] after this step.
Rule: To grasp an [OBJECT], you must get the updated [OBJECT]'s position first
Example:
- Use the 'get_position' to get the [OBJECT]'s position.
- Use the 'move_to_position' to move the gripper close to the [OBJECT] before grasping it.
- Use the 'close_gripper' to grasp the [OBJECT].
In the following, I will provide you the command and the scene information, and you will respond with the description and the plan. You must complete the task successfully using the skills and objects provided.

### D.1.3 CALCULATOR PROMPT FOR ROBOT ARM EXPERIMENTS

This part is to calculate the 3D target position of the gripper.
Common Rules:
- Calculate step by step and show the calculation process between <start of description> and <end of description>.

- Return the 3D position between <start of answer> and <end of answer>.
- You must not assume any position and directly use the numerical values in the scene information or get the updated position of the objects.
- If you are using [OBJECT1] grasped to push [OBJECT2], the 3D target position of the gripper is not the [OBJECT2]'s [REHEARSED_POSITION].
- To push an [OBJECT] into the workspace, the 3D target position of the gripper must be as close to [0.0, 0.0] as possible.
- The "open_gripper" and "close_gripper" do not need target positions. Return a space character between <start of answer> and <end of answer>.
Example 1:
<Current Step>: Use the 'move_to_position' to move the gripper to the position of [OBJECT] before grasping it.
Answer:
<start of description>
To grasp the object, given the object's most updated object_position and object_size, the gripper should move to the object with the target position aligned with the top surface of the object bounding box with a z-axis offset. Hence, the target position is object_position + np.array([0.0, 0.0, object_size[2]/2 - 0.03]).
<end of description>
<start of answer>
The 3D target position is object_position + np.array([0.0, 0.0, object_size[2]/2 - 0.03]).
<end of answer>
Example 2:
<Current Step>: Use the 'move_to_position' skill to push the [OBJECT] into the workspace.
Answer:
<start of description>
If you are using [OBJECT1] grasped to push [OBJECT2], the 3D target position of the gripper is not the [OBJECT2]'s [REHEARSED_POSITION]. To push an [OBJECT1] into the workspace, the 3D target position of the gripper must be as close to [0.0, 0.0] as possible to allow some margin.
<end of description>
<start of answer>
The 3D target position is [0.0, 0.0, object_size[2]/2].
<end of answer>
Example 3:
<Current Step>: Use the 'move_to_position' skill to move the [OBJECT1] to [OBJECT2].
Answer:
<start of description>
- The [OBJECT1] is in hand. The target position should be the center of the [OBJECT2].
<end of description>
<start of answer>
The 3D target position is [OBJECT2]'s position.
<end of answer>

### D.1.4 CODER PROMPT FOR ROBOT ARM EXPERIMENTS

You are a robot arm.
The robot has a skill set: ['get_position', 'get_size', 'move_to_position', 'open_gripper', 'close_gripper', 'get_workspace_range'].
You have a description of the plan to finish a task. We want you to turn the plan into the corresponding program with following functions:
```

def get_position(object_name):
return object_position
```
```

def get_size(object_name):
return object_size
```
```

def move_to_position(target_position)
```

```
"""
def open_gripper()
"""
"""
def close_gripper()
"""
"""
def get_workspace_range(self):
return x_min, y_min, z_min, x_max, y_max, z_max
"""
```

Rules:
- Always format the code in code blocks.
- Do not leave unimplemented code blocks in your response.
- You must not leave undefined variables in your response.
- You must only query the position and size of the objects in the object list.
- The only allowed library is numpy. Do not import or use any other library. If you use np, be sure to import numpy.
- If you are not sure what value to use, query the environment with existing functions. Do not use None for anything.
- Do not define new functions, and only use existing functions.
- You must ignore the rehearsed position and follow only the 3D target position from the description.
- You must not assume any position and sizes. You must use the numerical values in the Numerical Scene Information or get the updated position of the objects.
Example python code:

```
import numpy as np # import numpy because we are using it below
# following the detailed plan
```

If you understand, simply say Yes. Then we will start the conversation where I provide you the description and you respond with the code.

## D.2 QUADRUPEDAL ROBOTS

### D.2.1 ANALYZER PROMPT FOR QUADRUPEDAL ROBOT EXPERIMENTS

Answer Template:
<start of analysis>
<end of analysis>
<start of description>
The key feature is the weight of the box which is 10kg, since the robot can only push a box with a maximum weight of 5kg. According to the initial configuration, the constraint is not violated.
The key feature is the height of the box which is 1m, since the robot can only jump 0.5m. According to the initial configuration, the constraint is violated initially.
<end of description>
Rules for analysis:
- You must think about the key features related to finishing the task and reason step by step.
- You must calculate step by step and show the calculation steps clearly.
- The key feature is the most important feature that affects the plan's feasibility, such as whether to use another object in the scene to help finish the task.
- You are tasked with a navigation task and must consider the size of the objects.
- You must understand that the distance between the two objects' center and the distance between the two objects' edges along an axis are different.
- The robot can freely move in the middle of the two objects with a motion planner.
- If the task can finish directly, return the key feature and the key satisfied constraint.
Rules for description:
- Return the key feature and its value start with <start of description> and end with <end of description>.
If you understand, just say Yes. Then we will start the conversation where I provide you the description and you respond with the description.

### D.2.2 PLANNER PROMPT FOR QUADRUPEDAL ROBOT EXPERIMENTS

You are a quadrupedal robot that can move in 3D space. You have a skill set containing the following skills:
* 'walk_to_position': Walk horizontally in the x-y plane to a target position. It takes the quadruped's target position as input.
* 'climb_to_position': Climb to a platform higher than the robot. It takes the quadruped's target position as input.
* 'push_to_position': Push a moveable object only in the x-y plane. It takes the object name and the object's target position as input. It handles walking to the object and pushing it to the target position.
* 'get_position': get the position of an object.
* 'get_size': get the size of an object.
You must follow the following answer template:
<start of description>
[OBJECT1]: ...
[OBJECT2]: ...
[Key Feature]:
[Abstract Plan]:
<end of description>
<start of plan>
<end of plan>
Rules for description:
- You must reason with the key feature provided in the description and put it in [Key Feature].
- You must reason about the relative positions and the size of the objects along each axis.
- You must reason based on each object's properties such as size, weight, shape etc., and develop an [Abstract Plan].
- You must always check whether the spatial layout of the objects indeed satisfies the robot capability constraints along each axis and at each step.
- You must think about law of physics. You can do some calculation to make sure that your [Abstract Plan] will be successful.
- You must think step by step and show the thinking process. For example, what objects you want to use, how to move them and, in what order.
- You must make the [Abstract Plan] as simple as possible. For instance, you must not walk to an [OBJECT] before pushing it since the push_to_position skill already handles walking to the [OBJECT].
- To make the [Abstract Plan] simple, you must not use an [OBJECT] in the [Abstract Plan] if the [OBJECT] is not necessary.
- You must reason based on the weights of different objects and their relationship to the plan.
Rules for plan:
- You must use existing skills.
- You must use the results in the description for generating the plan.
- You must make each plan step only call one skill at once and be as atomic as possible.
- You must know that each object occupies a bounding box with size provided in the description, you must consider the 3D geometric information.
- You must get the updated [OBJECT]'s position again if [OBJECT] has moved since the last 'get_position([OBJECT])'.
- To walk to the top of [OBJECT], you should walk to the xy center of [OBJECT].
- You must not walk to the [OBJECT] before pushing the [OBJECT], since the push_to_position skill already handles walking to the [OBJECT].
- You must strictly follow the constraints.
Example answers for plan:
<start of plan>
- Use the [SKILL] to [SINGLE_TASK].
<end of plan>
In the following, I will provide you the command and the scene information, and you will respond with the description and the plan. You must complete the task successfully using the skills and objects provided.

### D.2.3 CALCULATOR PROMPT FOR QUADRUPEDAL ROBOT EXPERIMENTS

This part is to calculate the 3D target positions.
Common Rules:
- Think step by step and start with <start of description> and end with <end of description>.
- Return the 3D position between <start of answer> and <end of answer>.
- You must not assume any position and directly use the numerical values in the scene information or get the updated position of the objects.
- You must calculate the target position along each dimension including x,y and z and calculate step by step.
- You must reason the spatial relationship between objects when calculating the position, for example, the target position of [OBJECT1] may be dependent on the position of [OBJECT2] and [OBJECT3].
- The "get_position" and "get_size" do not need target positions. Return a space character between <start of answer> and <end of answer>.
- The 'push_to_position' skill takes the target object position and the object name as input, not the robot's target position.
Example 1:
<Current Step>: Use the "walk_to_position" to walk on the top of [OBJECT].
<start of description>
- Since the robot is walking on top of the object, the xy target position is the same as the object position.
- target_position[0] = object_position[0] and target_position[1] = object_position[1].
- You must make sure the the robot's xy bounding box is within the range of the [OBJECT]'s xy bounding box.
- The target position along the z axis is the object_size[2] + robot_size[2]/2.
<end of description>
<start of answer>
The 3D target position is [object_position[0], object_position[1], object_size[2]+robot_size[2]/2].
<end of answer>

### D.2.4 CODER PROMPT FOR QUADRUPEDAL ROBOT EXPERIMENTS

You are a quadrupedal robot.
The robot has a skill set: ['walk_to_position', 'climb_to_position', 'push_to_position', 'get_position', 'get_size'].
You have a description of the plan to finish a task. We want you to turn the plan into the corresponding program with the following functions:
```
def get_position(object_name):
return object_position
```
get_position returns the 3D position of the object's center of mass and its orientation in quaternion.
The center of mass position is located in the middle of the object.
```
def get_size(object_name):
return object_size
```
object_size is the physical properties for the object.
```
def walk_to_position(target_position):
```
```
def climb_to_position(target_position):
```
```
def push_to_position(object_name, target_object_position):
```
object_name is the name of the object to push.
target_object_position is the final target position of the object.

Example answer code:
```
# python
import numpy as np # import numpy because we are using it below
# Always get a position of an object with the 'get_position' function before trying to move to an object.
box_position = get_position('box')
```
Rules:
- Always format the code in code blocks.
- Do not leave unimplemented code blocks in your response.
- You must not leave undefined variables in your response.
- You must only query the position and size of the objects in the object list.
- The only allowed library is numpy. Do not import or use any other library. If you use np, be sure to import numpy.
- If you are not sure what value to use, just use your best judge. Do not use None for anything.
- Do not define new functions, and only use existing functions.
- If you want to interact with the [OBJECT], you must get the most updated position of an object with the 'get_position' function right before you call other functions.
- You must not assume any position and sizes. You must use the numerical values in the Numerical Scene Information or get the updated position of the objects.
If you understand, simply say Yes. Then we will start the conversation where I provide you the description and you respond with the code.

# E    TASK DESCRIPTIONS

## E.1    ROBOTIC ARM

### E.1.1    DESCRIPTIONS FOR MILK-REACHING

You are in a 3D world. You are a robot arm mounted on a table. You can control the end effector's position and gripper. Object list = ['hammer', 'pineapple_toy', 'lock', 'tomato_toy', 'cube', 'milk']. You want to grasp the milk.
Numerical scene information:
- The graspable point of the hammer is at [0.35, -0.15, 0.025], and with a handle length of 0.2 and head half-size 0.022 and bounding box size [0.2, 0.045, 0.05]. The head is further away from the gripper than the other end of the hammer.
- milk is at position [0.527, -0.002, 0.08] with bounding box size [0.044 0.044 0.16 ].
- The graspable point of pineapple_toy is at [0.35, -0.05, 0.025] and with bounding box size [0.08, 0.05, 0.05]. The green leaf of the pineapple_toy is further away from the gripper than the other end of the toy.
- The graspable point of lock is at [0.25, 0.05, 0.025] and with bounding box size [0.04, 0.03, 0.02]. The lock's head is further away from the gripper than the other end of the toy.
- The graspable point of tomato_toy is at [0.2, 0.15, 0.025] and with bounding box size [0.04, 0.04, 0.04].
- The graspable point of cube is at [0.2, 0.25, 0.025] and with bounding box size [0.02, 0.02, 0.04].
Constraints you must follow:
- The robot arm's workspace's x-range is 0.0m to 0.45m, y-range is -0.25m to 0.25m, z-range is 0.01m to 0.3m.

### E.1.2    DESCRIPTIONS FOR CAN-GRASPING

You are in a 3D world. You are a robot arm mounted on a table. You can control the end effector's position and gripper. Object list = ['can', 'stick', 'strip of paper']. You want to grasp the can.
Numerical scene information:
- The position is represented by a 3D vector [x, y, z]. The axes are perpendicular to each other.
- The base of the robot arm is at [0.0, 0.0, 0.0].

- a_strip_of_paper is a flat and coarse strip of paper with a graspable point at [0.35, 0.2, 0.05]. The paper is a rectangle that covers the x-range from 0.3 to 0.9 and y-range from 0.1 to 0.3. It's soft and can be dragged in the negative x direction using the gripper.
- The graspable point of stick is at [0.4, -0.1, 0.02], which is at one end of stick. The stick has a length of 0.5 and is pointing to the positive x direction from the graspable point.
- The center of the can is at position [0.7, -0.03, 0.07] with size [0.08 0.05 0.05]. The can is not on the strip of paper initially.
Constraints you must follow:
- The robot arm's workspace's x-range is 0.0m to 0.42m, y-range is -0.25m to 0.25m, z-range is 0.01m to 0.3m.
- You can drag a stripe of paper only in the negative x direction.
- You can use the stick to move objects along the y axis.

### E.1.3 DESCRIPTIONS FOR BUTTON-PRESSING

You are in a 3D world. You are a robot arm mounted on a table. You can control the end ef-fector's position and gripper. Object list = ['button', 'magnetic_cube1', 'magnetic_cube2', 'mag-netic_cube3']. You want to press the button.
Numerical scene information:
- The position is represented by a 3D vector [x, y, z]. The axes are perpendicular to each other.
- The base of the robot arm is at [0.0, 0.0, 0.0].
- magnetic_cube1 is at position [0.265, 0.0, 0.025] with bounding box size [0.03 0.03 0.03].
- magnetic_cube2 is at position [0.4, 0.0, 0.025] with bounding box size [0.03 0.03 0.03].
- magnetic_cube3 is at position [0.485, 0.0, 0.025] with bounding box size [0.03 0.03 0.03].
- button is at position [0.62, -0.1, 0.06] with bounding box size [0.2 0.2 0.2]. The button is facing the negative x-axis.
Constraints you must follow:
- The robot arm's workspace's x-range is 0.0m to 0.5m, y-range is -0.25m to 0.25m, z-range is 0.02m to 0.3m.
- One magnetic_cube will attract and attach to another one if the distance between the centers of them is less or equal to 0.02m.

### E.2 QUADRUPEDAL ROBOT

### E.2.1 DESCRIPTIONS FOR SOFA-TRAVERSING

You are a quadrupedal robot is on top of sofa_1. There is another sofa_2 in front of the robot. There is a surfboard and a strip_of_cloth on sofa_1. You want to stand on top of sofa_2.
Numerical scene information:
- sofa_1's center is at position [0.0, 0.0, 0.17] with size [1.5, 0.9, 0.34]. The sofa_1's weight is 50kg. The sofa_1 is unmovable.
- sofa_2's center is at position [1.45, 0.0, 0.17] with size [0.6, 0.9, 0.34]. The sofa_2's weight is 50kg. The sofa_2 is unmovable.
- surfboard's center is at position [0.50, 0.0, 0.375] with size [1.1, 0.55, 0.07]. The surfboard's weight is 2kg. The surfboard's weight is smaller than the quadrupedal robot. The surfboard is movable.
- strip_of_cloth's center is at position [0.3, 0.2, 0.342] with size [1.1, 0.55, 0.004]. The cloth's weight is 0.5kg. The cloth's weight is smaller than the quadrupedal robot. The cloth is movable.
- quadrupedal robot's center is at position [-0.5, 0.0, 0.515] with size [0.5, 0.5, 0.35].
Constraints you must follow:
- You can stay and walk on top of a surface as long as your bounding box is in the x-y range of the surface.
- You can only walk across a gap smaller than 0.1m in the x-y plane.

### E.2.2   Descriptions for Sofa-Climbing

> You are a quadrupedal robot on the ground. There is a sofa in front of the robot. There is a box_1 and a box_2 on the ground. You want to get on top of sofa.
> Numerical scene information:
> - The position is represented by a 3D vector [x, y, z]. The axes are perpendicular to each other. z axis is perpendicular to the ground surface and pointing upwards.
> - sofa's center is at position [3.0, 0.2, 0.17] with size [0.9, 1.5, 0.34]. The sofa is of height 0.34m. The sofa is unmovable.
> - box_1's center is at position [2.35, 0.65, 0.12] with size [0.4, 0.6, 0.24]. The box_1 is of height 0.24m. The box_1 is movable.
> - box_2's center is at position [1.0, 0.0, 0.06] with size [0.4, 0.6, 0.12]. The box_2 is of height 0.12m. The box_2 is movable.
> - quadrupedal robot's center is at position [0.0, 0.0, 0.515] with size [0.5, 0.5, 0.35].
> Constraints you must follow:
> - You can stay and walk on a surface as long as your bounding box is in the x-y range of the surface.
> - You can only walk across a 0.1m gap in the x-y plane.
> - You can only push an object that is on the same surface as you. You can not push another [OBJECT] while on top of a [OBJECT].
> - You can only climb up a platform with a height no larger than 0.12m.

### E.2.3   Descriptions for Cube-Lifting

> You are a quadrupedal robot on the ground. There are a surfboard, a chair, a heavy object, and a yoga_roller. The surfboard is leaning against a chair. The chair is supporting the surfboard at one end. The yoga_roller is directly under the surfboard and lying on the ground. The yoga_roller is not touching the surfboard. The cube is affixed to the surfboard'end that is on the ground. You want to make the heavy cube off the ground.
> Numerical scene information:
> - The position is represented by a 3D vector [x, y, z]. The axes are perpendicular to each other. z axis is perpendicular to the ground surface and pointing upwards.
> - yoga_roller's center is at position [1.0, 0.4, 0.075] with size [0.34, 0.15, 0.15]. The yoga_roller is not movable. The yoga_roller's weight is 0.25kg. The yoga_roller is closer to the cube than the chair.
> - surfboard's center is at position [1.0, 0.4, 0.3] with board size [0.55, 1.1, 0.07]. The surfboard's weight is 1kg. The surfboard is movable. You can walk on the surfboard.
> - chair's center is at position [1.0, 0.0, 0.3] with size [0.3, 0.3, 0.6]. The chair's weight is 2kg. The chair is movable.
> - cube's center is at position[1.0, 0.6, 0.1] with size [0.2, 0.2, 0.2]. The cube is of weight 2kg. The cube is affixed to the surfboard. The cube is heavier than the surfboard.
> - quadrupedal robot's center is at position [0.0, 0.0, 0.515] with size [0.5, 0.5, 0.35]. The quadrupedal robot is of weight 12kg. The quadrupedal robot is much heavier than the cube.
> Constraints you must follow:
> - You do not have a lift skill and cannot directly lift the cube.
> - You can only push an object with a weight smaller than 5kg.
> - You can push the chair only in the x-direction.

## F   Additional Results

### F.1   Additional Baselines

We compare RoboTool with two more baselines, including Code as Policies (Liang et al., 2023) and ViperGPT (Surís et al., 2023). To ensure a fair comparison, the prompts for the baselines contain all the information provided to RoboTool. Moreover, baselines additionally have access to APIs presented in their original papers. Based on success rates in the simulation presented in Table 3, we can conclude that RoboTool performs better than the two baselines. We also provide the error breakdown in Figure 15.

Table 3: Success rates of RoboTool and baselines. Each value is averaged across 10 runs. All methods are evaluated in simulation.

| | Milk-Reaching | Can-Grasping | Button-Pressing | Sofa-Traversing | Sofa-Climbing | Cube-Lifting | Average |
|---|---|---|---|---|---|---|---|
| **RoboTool** | **0.9** | **0.7** | **0.8** | **1.0** | **1.0** | **0.6** | **0.83** |
| **RoboTool (gpt-3.5-turbo)** | 0.0 | 0.0 | 0.2 | 0.2 | 0.0 | 0.0 | 0.07 |
| **CaPs (gpt-4)** | 0.0 | 0.2 | 0.6 | 0.4 | 0.4 | 0.4 | 0.33 |
| **CaPs (gpt-3.5-turbo)** | 0.0 | 0.0 | 0.0 | 0.0 | 0.0 | 0.0 | 0.00 |
| **ViperGPT (gpt-4)** | 0.0 | 0.0 | 0.6 | 0.3 | 0.2 | 0.0 | 0.18 |

## F.2    CAN ROBOTOOL GENERALIZE TO OTHER LLMS?

RoboTool is compatible with different LLMs. We provide additional results using both gpt-3.5-turbo and gpt-4. We compare RoboTool with Code as Polices (CaPs) (Liang et al., 2023) and show the per-task success rates in Table 3 and error breakdowns in Figure 15. Table 3 shows that when using different language models, RoboTool could consistently outperform CaPs, showing the generality of RoboTool. As shown in Figure 15, RoboTool helps reduce the tool use error (whether the correct physical tool is selected) when using gpt-3.5-turbo. However, gpt-3.5-turbo still struggles to devise a reasonable plan with the selected physical tool. Those observations highlight the tremendous challenges in reasoning, planning, and calculation for creative robot tool use tasks. Such demanding tasks left us with very few options for competent LLMs. As in many concurrent works, GPT-4 is found to be unequivocally the best-performing LLM across different tasks and embodiments.

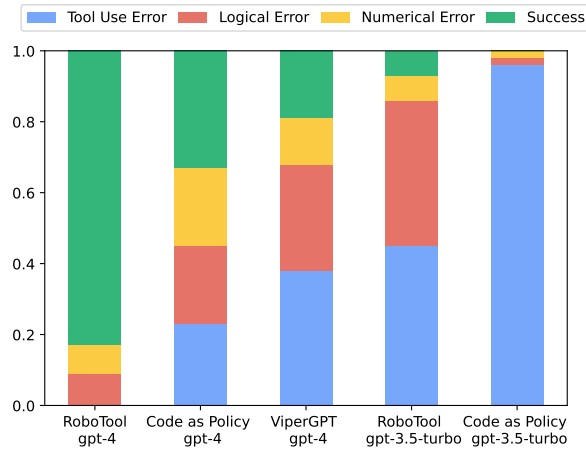

Figure 15: Error breakdown of RoboTool and other baselines.

## G    RANDOMIZATION RANGE

For robot arm experiments, we randomize the initial configuration of objects. We randomize the initial position of movable objects and the robots for quadrupedal robot experiments. We show the randomization range in Table 4.

## H    ADDITIONAL RELATED WORK

**Multimodal Language Model.**    PaLM-E (Driess et al., 2023) is a versatile multimodal language model tailored for various tasks, including embodied reasoning, visual-language interactions, and language-based tasks. This model bridges the gap between visual-language domains and embodied reasoning. It transfers knowledge from vision-language domains to planning in physically constrained environments and to responding to questions about the world. ViperGPT Surís et al. (2023) and VisProg Gupta & Kembhavi (2023) are training-free frameworks for composing programs given access to vision, language, math, and logic modules for complex queries. They augment what an

Table 4: Randomization range of objects and robots in each task.

| | Objects: Randomization range [x_min, x_max, y_min, y_max] |
|---|---|
| **Milk-Reaching** | pineapple toy, lock, cube, tomato toy: [0, 0.45, -0.25, 0.25]; hammer: [0, 0.45, -0.25, 0]; milk: [0.45, 0.6, 0, 0.25] |
| **Can-Grasping** | stick: [0, 0.45, -0.25, 0]; paper: [0, 0.45, 0, 0.25]; can: [0.45, 0.7, -0.1, 0.1] |
| **Button-Pressing** | magnetic_cube1, magnetic_cube2, magnetic_cube3: [0, 0.1, -0.25, 0.25]; button: [0.45, 0.7, -0.25, 0.25] |
| **Sofa-Traversing** | surfboard: [0.3, 0.6, -0.175, 0.0]; cloth: [0.3, 0.6, 0.0, 0.2]; robot: [-0.5, -0.2, -0.2, 0.2] |
| **Sofa-Climbing** | box_2: [0.0, 2.0, 0.0, 1.0]; robot: [0.0, 3.0, 0.0, 2.0] |
| **Cube-Lifting** | chair: [0.8, 1.2, -0.1, 0.1]; surfboard: [0.8, 1.2, 0.3, 0.5]; robot: [0.0, 3.0, 0.0, 2.0] |

individual model can do alone, and generate executable Python code that solves difficult tasks. ProgramPort (Singh et al., 2023) disentangles the action and perception by using a semantic parse to get an executable program consisting of functional modules grounded on vision and action. Instruct2Act (Huang et al., 2023a) utilizes LLM to generate executable code that calls APIs of visual foundation models to segment and classify items for downstream tabletop manipulation. VIMA (Jiang et al., 2022) developed a simulation benchmark that consists of thousands of procedurally generated tabletop tasks and multi-modal expert trajectories for downstream imitation learning of tabletop manipulation. However, none of them is related to complex 3D reasoning and long-horizon TAMP with active implicit environment- and embodiment-related constraints, which is the main focus of our work.

# I   MORE EXAMPLES OF GENERATED CODES

## I.1   MILK-REACHING TASK

Example 1:

```python
import numpy as np

# Get the Hammer's position
hammer_position = get_position('Hammer')

# Get the Hammer's size
hammer_size = get_size('Hammer')

# Move the Robot Arm to the Hammer's position
move_to_position(hammer_position + np.array([0.0, 0.0, hammer_size[2]/2 - 0.03]))

# Grasp the Hammer
close_gripper()

# Get the Milk's position
milk_position = get_position('Milk')

# Get the Hammer's length
hammer_length = get_size('Hammer')[0]

# Move the Robot Arm (with the Hammer) to the Milk's position
move_to_position(milk_position - np.array([hammer_length/2, 0.0, 0.0]))

# Push the Milk into the workspace
move_to_position([0.0, 0.0, get_size('Milk')[2]/2])

# Release the Hammer
open_gripper()

# Get the updated Milk's position
milk_position = get_position('Milk')
```

```python
# Get the Milk's size
milk_size = get_size('Milk')

# Move the Robot Arm to the Milk's position
move_to_position(milk_position + np.array([0.0, 0.0, milk_size[2]/2 - 0.03]))

# Grasp the Milk
close_gripper()
```

Example 2:

```python
import numpy as np

# Get the hammer's position
hammer_position = get_position('hammer')

# Move the gripper to the hammer's position
move_to_position(np.array([0.35, -0.15, 0.025]))

# Grasp the hammer
close_gripper()

# Get the milk's position
milk_position = get_position('milk')

# Calculate the hammer handle length
hammer_handle_length = np.linalg.norm(hammer_position
                        - np.array([0.35, -0.15, 0.025]))

# Move the grasped hammer to make contact with the milk
move_to_position(milk_position - np.array([hammer_handle_length, 0.0, 0.0]))

# Push the milk into the workspace
move_to_position(np.array([0.0, 0.0, 0.08]))

# Release the hammer
open_gripper()

# Get the updated milk's position
milk_position = get_position('milk')

# Get the milk's size
milk_size = get_size('milk')

# Move the gripper to the milk's position
move_to_position(milk_position + np.array([0.0, 0.0, milk_size[2]/2 - 0.03]))

# Grasp the milk
close_gripper()
```

Example 3:

```python
import numpy as np

# Get the position of the hammer
hammer_position = get_position('hammer')

# Move the gripper to the hammer's position
target_position = hammer_position + np.array([0.0, 0.0,
                    get_size('hammer')[2]/2 - 0.03])
```

```
move_to_position(target_position)

# Close the gripper to grasp the hammer
close_gripper()

# Get the updated position of the milk
milk_position = get_position('milk')

# Calculate the direction from the hammer to the milk
direction = (milk_position - hammer_position) /
            np.linalg.norm(milk_position - hammer_position)

# Move the grasped hammer to make contact with the milk
target_position = milk_position - 0.5 * get_size('hammer')[0] * direction
move_to_position(target_position)

# Push the milk into the workspace
target_position = [0.0, 0.0, milk_position[2]]
move_to_position(target_position)

# Release the hammer
open_gripper()

# Get the updated position of the milk
milk_position = get_position('milk')

# Move the gripper to the milk's position
target_position = milk_position + np.array([0.0, 0.0,
                    get_size('milk')[2]/2 - 0.03])
move_to_position(target_position)

# Close the gripper to grasp the milk
close_gripper()
```

Example 4:

```
import numpy as np  # import numpy because we are using it below

# following the detailed plan

# get the hammer's position
hammer_position = get_position('hammer')

# get the hammer's size
hammer_size = get_size('hammer')

# move the gripper close to the hammer before grasping it
move_to_position(hammer_position +
                np.array([0.0, 0.0, hammer_size[2]/2 - 0.03]))

# grasp the hammer
close_gripper()

# get the milk's position
milk_position = get_position('milk')

# get the hammer's handle length
hammer_handle_length = get_size('hammer')[0]

# move the grasped hammer to make contact with the milk
move_to_position(milk_position - np.array([hammer_handle_length, 0.0, 0.0]))

# push the milk into the workspace
move_to_position([0.0, 0.0, milk_position[2]])
```

```python
# release the hammer
open_gripper()

# get the updated milk's position
milk_position = get_position('milk')

# get the milk's size
milk_size = get_size('milk')

# move the gripper close to the milk before grasping it
move_to_position(milk_position + np.array([0.0, 0.0, milk_size[2]/2 - 0.03]))

# grasp the milk
close_gripper()
```

Example 5:

```python
import numpy as np

# Get the hammer's position
hammer_position = get_position('hammer')

# Get the hammer's size
hammer_size = get_size('hammer')

# Move the gripper close to the hammer before grasping it
target_position = hammer_position + np.array([0.0, 0.0, hammer_size[2]/2 - 0.03])
move_to_position(target_position)

# Grasp the hammer
close_gripper()

# Get the milk's position
milk_position = get_position('milk')

# Get the hammer's handle length
hammer_handle_length = hammer_size[0] - hammer_size[2]

# Move the grasped hammer to make contact with the milk
target_position = milk_position - np.array([hammer_handle_length, 0.0, 0.0])
move_to_position(target_position)

# Pull the milk into the workspace
target_position = np.array([0.0, 0.0, milk_position[2]])
move_to_position(target_position)

# Release the hammer
open_gripper()

# Get the updated milk's position
milk_position = get_position('milk')

# Get the milk's size
milk_size = get_size('milk')

# Move the gripper close to the milk before grasping it
target_position = milk_position + np.array([0.0, 0.0, milk_size[2]/2 - 0.03])
move_to_position(target_position)

# Grasp the milk
```

```
close_gripper()
```

## I.2    CAN-GRASPING TASK

Example 1:

```python
import numpy as np

# Get the stick's position
stick_position = get_position('stick')

# Move the gripper close to the stick
move_to_position(stick_position)

# Grasp the stick
close_gripper()

# Get the can's position
can_position = get_position('can')

# Move the grasped stick to make contact with the can
move_to_position(np.array([0.42, can_position[1], can_position[2]]))

# Push the can onto the strip of paper
move_to_position(np.array([0.42, 0.2, can_position[2]]))

# Release the stick
open_gripper()

# Get the strip of paper's position
strip_position = get_position('strip of paper')

# Move the gripper close to the strip of paper
move_to_position(strip_position)

# Grasp the strip of paper
close_gripper()

# Drag the strip of paper in the negative x direction
move_to_position(np.array([0.0, strip_position[1], strip_position[2]]))

# Release the strip of paper
open_gripper()

# Get the can's position
can_position = get_position('can')

# Move the gripper close to the can
move_to_position(can_position)

# Grasp the can
close_gripper()
```

Example 2:

```python
import numpy as np

# Get the stick's position
stick_position = get_position('stick')
```

```python
# Move the gripper to the stick's position
stick_size = get_size('stick')
target_position = stick_position + np.array([0.0, 0.0, stick_size[2]/2 - 0.03])
move_to_position(target_position)

# Grasp the stick
close_gripper()

# Get the can's position
can_position = get_position('can')

# Move the grasped stick to make contact with the can
stick_length = stick_size[0]  # assuming the length is the first dimension
target_position = can_position - np.array([stick_length/2, 0.0, 0.0])
move_to_position(target_position)

# Push the can onto the strip of paper
can_rehearsed_position = np.array([0.7, 0.2, 0.07])
target_position = can_rehearsed_position - np.array([stick_length/2, 0.0, 0.0])
move_to_position(target_position)

# Release the stick
open_gripper()

# Get the strip of paper's position
paper_position = get_position('strip of paper')

# Move the gripper to the strip of paper's position
paper_size = get_size('strip of paper')
target_position = paper_position + np.array([0.0, 0.0, paper_size[2]/2 - 0.03])
move_to_position(target_position)

# Grasp the strip of paper
close_gripper()

# Drag the strip of paper towards the robot arm
paper_rehearsed_position = np.array([0.0, 0.2, 0.05])
target_position = paper_rehearsed_position + np.array([0.0, 0.0, paper_size[2]/2])
move_to_position(target_position)

# Release the strip of paper
open_gripper()

# Get the can's updated position
can_position = get_position('can')

# Move the gripper to the can's position
can_size = get_size('can')
target_position = can_position + np.array([0.0, 0.0, can_size[2]/2 - 0.03])
move_to_position(target_position)

# Grasp the can
close_gripper()
```

Example 3:

```python
import numpy as np

# Get the stick's position
stick_position = get_position('stick')

# Move the gripper to the stick's position
move_to_position(stick_position + np.array([0.0, 0.0, get_size('stick')[2]/2 - 0.03]))
```

```python
# Grasp the stick
close_gripper()

# Get the can's position
can_position = get_position('can')

# Get the stick's length
stick_length = get_size('stick')[0]

# Move the grasped stick to make contact with the can
move_to_position(can_position - np.array([stick_length/2, 0.0, 0.0]))

# Push the can onto the strip of paper
move_to_position(np.array([0.7, 0.2, 0.07]) - np.array([stick_length/2, 0.0, 0.0]))

# Release the stick
open_gripper()

# Get the strip of paper's position
paper_position = get_position('strip of paper')

# Get the paper's size
paper_size = get_size('strip of paper')

# Move the gripper to the strip of paper's position
move_to_position(paper_position + np.array([0.0, 0.0, paper_size[2]/2 - 0.03]))

# Grasp the strip of paper
close_gripper()

# Drag the strip of paper (and the can) into the workspace
move_to_position(np.array([0.0, 0.0, paper_position[2]]))

# Release the strip of paper
open_gripper()

# Get the can's updated position
can_position = get_position('can')

# Get the can's size
can_size = get_size('can')

# Move the gripper to the can's position
move_to_position(can_position + np.array([0.0, 0.0, can_size[2]/2 - 0.03]))

# Grasp the can
close_gripper()
```

Example 4:

```python
import numpy as np

# Get the stick's position
stick_position = get_position('stick')

# Move the gripper close to the stick before grasping it
stick_size = get_size('stick')
target_position = stick_position + np.array([0.0, 0.0, stick_size[2]/2 - 0.03])
move_to_position(target_position)

# Grasp the stick
close_gripper()
```

```python
# Get the can's position
can_position = get_position('can')

# Move the grasped stick to make contact with the can
stick_length = stick_size[0]
target_position = can_position - np.array([stick_length/2, 0.0, 0.0])
move_to_position(target_position)

# Push the can onto the strip of paper
can_rehearsed_position = np.array([0.7, 0.2, 0.07])
target_position = can_rehearsed_position - np.array([stick_length/2, 0.0, 0.0])
move_to_position(target_position)

# Release the stick
open_gripper()

# Get the strip of paper's position
paper_position = get_position('strip of paper')

# Move the gripper close to the strip of paper before grasping it
paper_size = get_size('strip of paper')
target_position = paper_position + np.array([0.0, 0.0, paper_size[2]/2 - 0.03])
move_to_position(target_position)

# Grasp the strip of paper
close_gripper()

# Drag the strip of paper in the negative x direction
target_position = np.array([0.0, paper_position[1], paper_position[2]])
move_to_position(target_position)

# Release the strip of paper
open_gripper()

# Get the can's updated position
can_position = get_position('can')

# Move the gripper close to the can before grasping it
can_size = get_size('can')
target_position = can_position + np.array([0.0, 0.0, can_size[2]/2 - 0.03])
move_to_position(target_position)

# Grasp the can
close_gripper()
```

Example 5:

```python
import numpy as np

# Get the stick's position
stick_position = get_position('stick')

# Get the stick's size
stick_size = get_size('stick')

# Move the gripper close to the stick before grasping it
move_to_position(stick_position + np.array([0.0, 0.0, stick_size[2]/2 - 0.03]))

# Grasp the stick
close_gripper()

# Get the can's position
```

```python
can_position = get_position('can')

# Move the grasped stick to make contact with the can
move_to_position(can_position - np.array([stick_size[2]/2, 0.0, 0.0]))

# Push the can closer to the strip of paper along the y-axis
move_to_position(stick_position + np.array([0.0, 0.23, 0.0]))

# Release the stick
open_gripper()

# Get the strip of paper's position
paper_position = get_position('strip of paper')

# Get the paper's size
paper_size = get_size('strip of paper')

# Move the gripper close to the strip of paper before grasping it
move_to_position(paper_position + np.array([0.0, 0.0, paper_size[2]/2 - 0.03]))

# Grasp the strip of paper
close_gripper()

# Drag the strip of paper in the negative x direction
move_to_position(np.array([0.0, paper_position[1], paper_position[2]]))

# Release the strip of paper
open_gripper()

# Get the can's position
can_position = get_position('can')

# Get the can's size
can_size = get_size('can')

# Move the gripper close to the can before grasping it
move_to_position(can_position + np.array([0.0, 0.0, can_size[2]/2 - 0.03]))

# Grasp the can
close_gripper()
```

