# OpenReview forum: "Creative Robot Tool Use with Large Language Models"
_ICLR.cc/2024/Conference — Submitted to ICLR 2024_

### Official Review · Reviewer_xccj · 2023-10-26

**Soundness:** 2 fair
**Presentation:** 4 excellent
**Contribution:** 2 fair
**Rating:** 5
**Confidence:** 4

**Summary:**

The paper explores the interesting question of enabling robots to use tools, taking into account constraints from both the robot and its environment. The authors propose the RoboTool system, which augments the coder module with additional analyzer, planner, and calculator modules. This paper presents a benchmark encompassing three tool-usage categories: tool selection, sequential tool usage, and tool manufacturing, evaluated across two types of robots. Through carefully designed experiments, the authors show that their system exhibits innovative tool use.

**Strengths:**

* Leveraging LLMs to delve into robot tool usage is a compelling approach. The inherent common sense knowledge within LLMs may offer invaluable insights to the robot's tool utilization process.
* The RoboTool System builds on the prior concept of LLM code generation (framed as code-as-policies) and merges it with new analyzer, calculator, and planner modules. This integration aids in breaking down the task, enabling the LLM to more effectively suggest beneficial solutions. The ablation studies provide evidence of the effectiveness of these newly introduced modules.
* The categorization in the benchmark is well-designed as a starting point to explore the robot tool usage with LLMs.

**Weaknesses:**

* Although this paper focus on high-level planning using tools, the provided descriptions are too detailed on the targetted tools, which makes it hard to see if the hints make LLM propose the solutions. For example, for the Milk-Reaching example, the hammer is provided with detailed instructions on how to grasp, the descriptions on its layout, while other objects are not described that detailedly. Such bias can make the results unfair. And in all 6 experiments, the number of objects in the descriptions are limited, it may be not hard for the LLM to pick a related object.
* The benchmark only contains 6 demos, with limited diversity on the layout of the objects. For example, for the milk-reaching demo, the hammer is always in the correct direction. With similar description, actually the hammer can be in multiple potential directions, which will definitely influence the planning the success rate of the task. Such challenging examples are not considered in the constructed benchmarks. And with the natural language description, it cannot avoid the limitation to describe the 3D world. Without access to the full information, it’s hard to imagine the performance of the system on complicated tasks. Need to show more results on the robustness for the system on various examples.
* The descriptions are sometimes confusing. For example, in the Cube-Lifting, the cube weight is 10kg, and the robot weight is also 10kg, then in the video, why the robot will fall down so quickly when it goes to another side. It’s a bit confusing if the description reflects the real property and why not use the real physical attributes.
* For some constraints mentioned in the description, it’s unclear why the constraints make sense. For example, in the Cube-Lifting example, in the constraints, “you can push the chair only in the x-direction”, it makes readers confusing if the input description is well-tuned for the specific example and how such things make the system generalizable across different tasks.

**Questions:**

* Regarding the benchmark, are there additional results showcasing varying object layouts for each demonstration while maintaining a consistent description format?
* In the demos, how does the system respond when constraints in the input description are removed? How sensitive is it to changes in the description?
* Why not for all objects, give the same set of attributes no matter if the attributes are useful enough? In this way, it can better show if the system is able to really extract the useful information from the descriptions without hints.
* Although this work try demonstrating the tool usage ability in the high-level setting, it’s hard to ignore the influence from different details. How to make sure the description describe the scene without heavy human designing?
* How to make sure the given grasping point and other attributes or additional description for one object are not hints to the tool usage demo?

---

> ### Author Response · Authors · 2023-11-17
> **Response to Reviewer xccj (Part 1)**
>
> Thank you for the detailed and insightful reviews! We appreciate that you found leveraging LLMs for creative tool use is a compelling approach, our method is effective, and the benchmark is well-designed. Here are our responses to your concerns:
>
> > Q1: Limited task diversity.
>
>  We want to highlight that designing six creative tool-use tasks and their variants to show the discriminative tool-use behaviors contributes to the robotics community, as discussed in [1]. Unlike many robotic manipulation tasks inspired by household activities, creative tool use is more like solving intricate physical puzzles with many design thoughts injected into the process. RoboTool is the first step towards designing a standard robot tool use benchmark. We hope our research could inspire more work on this critical yet underexplored topic.
>
> **Ref:**
>
> [1] Qin, Meiying, Jake Brawer, and Brian Scassellati. 2022. “Robot Tool Use: A Survey.” Frontiers in Robotics and AI 9: 1009488.
>
>
> > Q1.1: Are the object layouts changed?
>
> Thanks for the question. The positions are indeed randomized. For robot arm experiments, we randomize the initial configuration of objects. We randomize the initial position of movable objects and the robots for quadrupedal robot experiments. We show the randomization ranges in Table 1 and add it to Appendix G in the updated draft.
>
> Table 1: Randomization of objects in each task.
> | | Objects: Randomization range [x_min, x_max, y_min, y_max] |
> | ------------- | ------- |
> | Milk-Reaching | pineapple toy, lock, cube, tomato toy: [0, 0.45, -0.25, 0.25]; hammer: [0, 0.45, -0.25, 0]; milk: [0.45, 0.6, 0, 0.25]  |
> | Can-Grasping | stick: [0, 0.45, -0.25, 0]; paper: [0, 0.45, 0, 0.25]; can: [0.45, 0.7, -0.1, 0.1]  |
> | Button-Pressing | magnetic_cube1, magnetic_cube2, magnetic_cube3: [0, 0.1, -0.25, 0.25]; button: [0.45, 0.7, -0.25, 0.25]  |
> | Sofa-Traversing | surfboard: [0.3, 0.6, -0.175, 0.0]; cloth: [0.3, 0.6, 0.0, 0.2]; robot: [-0.5, -0.2, -0.2, 0.2];   |
> | Sofa-Climbing | box_2: [0.0, 2.0, 0.0, 1.0]; robot: [0.0, 3.0, 0.0, 2.0];   |
> | Cube-Lifting | chair: [0.8, 1.2, -0.1, 0.1]; surfboard: [0.8, 1.2, 0.3, 0.5]; robot: [0.0, 3.0, 0.0, 2.0];   |
>
> > Q1.2: In milk-reaching experiment, the number of objects in the descriptions is limited. It may not be hard for the LLM to pick a related object.
>   - We increase the number of objects that may serve as tools to five, including a hammer, a pineapple toy, a lock, a tomato toy, and a cube. We also provide the same amount of information for each object, including (1) graspable points for non-rectangular objects and the position of rectangular objects and (2) bounding box sizes of each object. Considering the non-rectangular shape of the hammer and pineapple toy, we added a description related to the orientation. However, the orientation description does not affect the tool selection in experiments. After rerunning experiments, we noticed that RoboTool can still correctly select the hammer as the tool among ten runs.
>  - Besides selecting the correct tool, how and when to use the tool are also important. For example, the Can-Grasping task has three objects, five skills (2 have a discrete action parameter, 1 has continuous action parameters), and 15 planning steps. The underlying complexity of planning is significant, and any errors in the intermediate steps could lead to a catastrophic failure. Despite the difficulties, RoboTool achieved a high success rate across different domains, indicating it is not just randomly picking an object to interact with.

---

> > ### Author Response · Authors · 2023-11-17
> > **Response to Reviewer xccj (Part 2)**
> >
> > > Q1.3: In milk-reaching experiments, the hammer is always pointing in the same direction.
> >  - We only provide a skill to move the gripper in the 3D translational space. If the hammer is pointing in the opposite direction, the robot cannot complete the task since it does not have a rotational skill. This is a limitation of our action primitives rather than the general framework. Please note that RoboTool is compatible with different skills and diverse skills will enable more capable agents to handle various task configurations robustly.
> >  - As discussed in the limitation section and suggested by the reviewer, multi-modality is a promising future direction. However, the current version of SOTA VLM models are insufficient in 3D reasoning [2, 3]. Moreover, existing VLMs still cannot get the numerical scene descriptions (with positions and sizes) and the constraints directly without leveraging parsing APIs. We follow the setting in VoxPoser [4] to get the position and sizes of objects through OWL-ViT [5]. Future stronger vision foundation models will further enhance RoboTool's cognition ability in object affordance, robustness to online disturbance, etc.
> >
> > **Ref:**
> >
> > [2] Liu, Fangyu, Guy Emerson, and Nigel Collier. "Visual spatial reasoning." Transactions of the Association for Computational Linguistics 11 (2023): 635-651.
> >
> > [3] Yamada, Yutaro, et al. "Evaluating Spatial Understanding of Large Language Models." arXiv preprint arXiv:2310.14540 (2023).
> >
> > [4] Huang, Wenlong, et al. "Voxposer: Composable 3d value maps for robotic manipulation with language models." arXiv preprint arXiv:2307.05973 (2023).
> >
> > [5] Minderer, Matthias, et al. "Simple open-vocabulary object detection." European Conference on Computer Vision. Cham: Springer Nature Switzerland, 2022.
> >
> >
> >
> > > Q2: Description for constraints, e.g., 'you can push the chair only in the x-direction'. Are they important? Could they be removed?
> >
> >  - Without this constraint, RoboTool still produces the same high-level plan skeleton, but with a difference in the target position to push the chair. If the chair is pushed in the y-direction, the robot may clash with the setup, and the surfboard will fall on top of the robot and fail the task. This is related to the realistic setup and the embodiment of the robot rather than our general framework.
> >  - Correct constraints are critical to success in such challenging tasks. Traditional TAMP literature encodes kinematics and dynamics constraints as predicates and logical rules. For example, in [6], in order to impose constraints on the action primitive ```grasp(X, Y)```, ```[inside X Y] (staFree X Y)``` refers to be predefined, where ```[inside X Y]``` defines point ```X``` is inside object ```Y```, and ```(staFree X Y)``` refers stable (constrained to zero velocity) free joints from ```X``` to ```Y```. In cognitive robotics, there is a rich body of work that requires identifying, modeling, and learning constraints, for example, learning constraints from human feedback [7]. Inputing constraints is not unique to our method but rather common in the robotics literature. Similar to [8, 9], the user provides free-form natural language rather than demonstrations or predefined predicates and logical rules.
> >  - With that being said, beyond explicit language descriptions, we think there are indeed more seamless ways to obtain such knowledge through interactions and encode them, such as physically grounded Vision-Language Models [10]. However, there are currently no VLMs that can directly provide grounded constraints information. Nevertheless, RoboTool is one of the first steps towards exploring the environment- and embodiment-related constraints and incorporating them into decision-making.
> >
> >
> > **Ref:**
> >
> > [6] Toussaint, Marc A., et al. "Differentiable physics and stable modes for tool-use and manipulation planning." (2018).
> >
> > [7] Fitzgerald, Tesca, Ashok Goel, and Andrea Thomaz. "Modeling and learning constraints for creative tool use." Frontiers in Robotics and AI 8 (2021): 674292.
> >
> > [8] Stepputtis, Simon, et al. "Language-conditioned imitation learning for robot manipulation tasks." Advances in Neural Information Processing Systems 33 (2020): 13139-13150.
> >
> > [9] Sharma, Pratyusha, et al. "Correcting robot plans with natural language feedback." Robotics: Science and Systems 2022.
> >
> > [10] Gao, Jensen, et al. "Physically Grounded Vision-Language Models for Robotic Manipulation." arXiv preprint arXiv:2309.02561 (2023).

---

> ### Author Response · Authors · 2023-11-17
> **Response to Reviewer xccj (Part 3)**
>
> > Q3: You should give the same set of attributes to all objects in the tasks:
>  - Thanks for the suggestion. We have cleaned up all the task descriptions. For robot arm experiments, the task description contains (1) graspable points for non-rectangular objects and the position of rectangular objects and (2) bounding box sizes of each object. For quadrupedal robot experiments, the task description contains each object's center position and size. Since tasks in quadrupedal robot experiments are loco-manipulation tasks, we also include the weight of each object in the scene.
>      - **Imbalance description in Milk-Reaching; Inaccurate cube weight of Cube-Lifting.** Thanks for the insightful question. We have cleaned up the object description and changed the cube weight to the actual value.
>  - After rerunning the experiments, we noticed that RoboTool has a slight performance drop (0.87 -> 0.83), but it is insignificant.
>
>
>
>
> > Q4 & Q5: How to make sure the description describes the scene without heavy human design? Does the description hint at the final plan?
>  - We use human-designed templates due to the limitation of existing Vision Language Models. As mentioned in the response to Q1.3, the SOTA VLM models are insufficient in 3D reasoning, and it is hard to get a reasonable estimate of object positions and sizes directly without the help of existing APIs. Moreover, our tasks require information about the constraints and weights, which are hard to get from VLMs. Hence, to explore the complex creative tool-use tasks with existing foundation models, we have no choice but to provide a template description.
>  - Based on the reviewer's suggestion in Q4, we have cleaned up the description and made sure that different objects in the scene are provided with similar amounts of information, such as positions, sizes, and weights. Please note that the provided information is sufficient but may be unnecessary for finishing the tasks. In other words, we have provided distractors, such as other objects in tool-selection tasks, which require LLMs to filter the information and reason based on the most important information.
>  - Even with the same amount of information, the state-of-the-art baselines still have low success rates, as shown in Table 2. RoboTool performs better than baselines by a large margin, highlighting the task complexity of creative tool-use problems.
>
> Table 2: Success rates of RoboTool and additional baselines.
> |  Task | RoboTool (ours) | CaPs | ViperGPT |
> | --- | -------- | -------------- | -------- |
> |  Milk-Reaching  | 0.9 | 0.0 | 0.0 |
> |  Can-Grasping  | 0.7 | 0.2 | 0.0 |
> |  Button-Pressing  | 0.8 | 0.6 | 0.6 |
> |  Sofa-Traversing  | 1.0 | 0.4 | 0.3 |
> |  Sofa-Climbing  | 1.0 | 0.4 | 0.2 |
> |  Cube-LIfting  | 0.6 | 0.4 | 0.0 |
> |  Average  | 0.83 | 0.33 | 0.18 |

---

> ### Author Response · Authors · 2023-11-20
> **Looking forward to your rebuttal feedback!**
>
> Thank you for the constructive feedback and suggestions! As the discussion period is ending, we would appreciate you kindly checking our response. Please do not hesitate to contact us if there are other clarifications we can offer. Thanks!

---

> ### Author Response · Authors · 2023-11-22
> **Followup on the rebuttal as the discussion period is ending soon**
>
> Dear reviewer xccj,
>
> With the rebuttal period ending tomorrow, we would like to ask whether our response addressed your questions and alleviated your concerns. If so, could you please kindly consider raising the score?
>
> Best,
>
> Authors

---

### Official Review · Reviewer_b6vg · 2023-10-31

**Soundness:** 3 good
**Presentation:** 3 good
**Contribution:** 2 fair
**Rating:** 1
**Confidence:** 4

**Summary:**

This paper presents RoboTool, a method for enabling tool use in robots using large language models (LLMs). Besides this prompt-based task and motion planning framework, the paper also proposes a benchmark of 6 tool use tasks evaluating tool selection, sequential tool use, and tool manufacturing capabilities. Tasks involve a robotic arm and a quadrupedal robot. Experiments in simulation and the real world demonstrate that RoboTool can successfully accomplish the tool use tasks.

**Strengths:**

1. Leveraging the recent wealth of LLM research for improving robotics is a highly desirable research direction that is well-explored in this paper.

**Weaknesses:**

1. Experiments are weak: in particular, the authors propose a new benchmark, but only evaluate their method on it. To ascertain the value of the benchmark suite, additional baselines need to be included. To assess the strength of contributions of this "learning-free" approach, it should be run on existing, standardized benchmarks, such as those included in [3] or [6].

2. Lack of novelty: works such as [1], [2], [3], [4] and [5] have taken similar approaches to neuro-symbolic learning and robotic manipulation, via LLM-generated programs or TAMP structures. Moreover, it's not particularly satisfying to me that the entire method interacts only with GPT-4 at the API level. The paper in effect becomes a "prompt engineering" work, which, while interesting, does not meet the bar for original technical contribution at ICLR.

[1] [Code as Policies: Language Model Programs for Embodied Control](https://arxiv.org/abs/2209.07753)

[2] [ViperGPT: Visual Inference via Python Execution for Reasoning](https://arxiv.org/abs/2303.08128)

[3] [Programmatically Grounded, Compositionally Generalizable Robotic Manipulation](https://arxiv.org/abs/2304.13826)

[4] [Visual Programming: Compositional visual reasoning without training](https://arxiv.org/abs/2211.11559)

[5] [Instruct2Act: Mapping Multi-modality Instructions to Robotic Actions with Large Language Model](https://arxiv.org/abs/2305.11176)

[6] [VIMA: General Robot Manipulation with Multimodal Prompts](https://arxiv.org/abs/2210.03094)

**Questions:**

1. The ablations provided are interesting and welcome, but could the authors include some more well-established baselines such as [1] or [2] in this evaluation?

2. Can the authors address why an API-only algorithm is sufficiently novel? In particular, I don't see where there is any learning of representations, which nominally is what ICLR is focused on.

[1] [Code as Policies: Language Model Programs for Embodied Control](https://arxiv.org/abs/2209.07753)

[2] [ViperGPT: Visual Inference via Python Execution for Reasoning](https://arxiv.org/abs/2303.08128)

---

> ### Author Response · Authors · 2023-11-17
> **Response to Reviewer b6vg (Part 1)**
>
> Thank you for the valuable feedback. We appreciate that you found leveraging LLMs for creative tool use is a highly desirable research direction and well-explored in this paper. Here is our response regarding concerns about novelty, baselines, and benchmarks.
>
> > Novelty of the paper.
>
> We respectfully disagree with the reviewer about the novelty of RoboTool. Here is a summary of our contributions:
>  - **Novel Problem:** Creative tool use is an important yet under-explored problem in robotics. It will greatly expand the capability of robots to solve tasks that are impossible originally.
>  - **Novel Benchmark:** We designed a novel challenging benchmark to test specifically the creative tool use behavior, which contains two robot embodiments and three categories of creative tool use. Such a benchmark is highly needed and non-trivial to design [1].
>  - **Novel Method:** Although the planner-coder LLM agent has been explored in existing literature, blindly using existing API-based methods such as Code as Policy fails to address the challenges posed by creative tool use in terms of cognition, reasoning, and planning. As shown below, the suggested baselines failed to complete any of the tasks in our benchmark with a high success rate.
>
> **Ref:**
>
> [1] Qin, Meiying, Jake Brawer, and Brian Scassellati. 2022. “Robot Tool Use: A Survey.” Frontiers in Robotics and AI 9: 1009488.
>
> > ICLR is focused on representation learning, while RoboTool is a training-free method.
>
>  - Despite the name, ICLR is about more than just learning representation, as documented by the non-comprehensive topic list on the [official website](https://iclr.cc/Conferences/2024/CallForPapers). This year in ICLR, there are at least 90 submissions related to training-free LLM Agents. When looking at the accepted papers of this year NeurIPS, one can find at least 20 impactful training-free API-based LLM agents, such as HuggingGPT [2], Reflexion [3], DEPS [4], AVIS [5], CAAFE [6], and SheetCopilot [7]. With increasingly capable LLMs, it is critical to explore what they can or cannot do, which is underexplored especially in the physical interaction and 3D reasoning domains. RoboTool is one of the first steps towards reasoning implicit environment- and embodiment-constraints and long-horizon hybrid discrete-continuous planning. Therefore, RoboTool is well-suited to the top AI publication venues.
>
> **Ref:**
>
> [2] Shen, Yongliang, et al. "Hugginggpt: Solving ai tasks with chatgpt and its friends in huggingface." NeurIPS 2023.
>
> [3] Shinn, Noah, Beck Labash, and Ashwin Gopinath. "Reflexion: an autonomous agent with dynamic memory and self-reflection." NeurIPS 2023.
>
> [4] Wang, Zihao, et al. "Describe, explain, plan and select: Interactive planning with large language models enables open-world multi-task agents." NeurIPS 2023.
>
> [5] Hu, Ziniu, et al. "AVIS: Autonomous Visual Information Seeking with Large Language Models." NeurIPS 2023.
>
> [6] Hollmann, Noah, Samuel Müller, and Frank Hutter. "GPT for Semi-Automated Data Science: Introducing CAAFE for Context-Aware Automated Feature Engineering." NeurIPS 2023.
>
> [7] Li, Hongxin, et al. "SheetCopilot: Bringing Software Productivity to the Next Level through Large Language Models." arXiv preprint arXiv:2305.19308 (2023).

---

> > ### Author Response · Authors · 2023-11-17
> > **Response to Reviewer b6vg (Part 2)**
> >
> > > Additional baselines
> >
> >  - Thanks for pointing us to the baselines. We compare our proposed RoboTool with Code as Policies (CaPs) and ViperGPT. To ensure a fair comparison, the prompts for the baselines contain all the information provided to RoboTool. Moreover, baselines additionally have access to APIs presented in their original papers. All the baselines and RoboTool are based on gpt-4. We show the success rates in the simulation in the following Table 1. The results show that RoboTool performs better than the two baselines. We also provide the error breakdown in Table 2 below. Our results show that RoboTool can successfully reduce the physical tool-use errors (which physical tool to use), logical errors (how to devise a plan), and numerical errors (what are the parameters for each robot skill). The results strongly support the contributions of RoboTool.
> >
> > Table 1: Success rates of RoboTool and additional baselines.
> > |  Task | RoboTool (ours) | CaPs | ViperGPT |
> > | --- | -------- | -------------- | -------- |
> > |  Milk-Reaching  | 0.9 | 0.0 | 0.0 |
> > |  Can-Grasping  | 0.7 | 0.2 | 0.0 |
> > |  Button-Pressing  | 0.8 | 0.6 | 0.6 |
> > |  Sofa-Traversing  | 1.0 | 0.4 | 0.3 |
> > |  Sofa-Climbing  | 1.0 | 0.4 | 0.2 |
> > |  Cube-LIfting  | 0.6 | 0.4 | 0.0 |
> > |  Average  | 0.83 | 0.33 | 0.18 |
> >
> > Table 2: Error beakdowns for the additional baselines.
> > |     | RoboTool | CaPs | ViperGPT |
> > | --- | ---------------- | ------------ | -------------------- |
> > Tool Use Error| 0.00 |  0.23| 0.38|
> > Logical Error| 0.09 |  0.22 | 0.30|
> > Numerical Error| 0.08|  0.22 | 0.13|
> > Success| 0.83| 0.33| 0.19 |
> >
> > > Additonal benchmarks PROGRAMPORT [3] and VIMA [6].
> >  - Once again, we are focusing on the novel creative tool-use problem that is underexplored in the existing literature. There is no pre-existing standard benchmark to test the creative tool-use capability. One of our contributions is that we are the first ones to design a benchmark specifically for such a purpose.
> >  - Moreover, the suggested benchmarks are focusing on very different problems. For example, [3] and [6] focused on pick-and-place manipulation in a 2D table-top imitation learning setting. None of them is related to complex 3D reasoning and long-horizon TAMP with activated environment- and embodiment-related constraints, which is the main focus of our work. Therefore, testing RoboTool on those benchmarks provides limited insights into RoboTool.

---

> ### Author Response · Authors · 2023-11-20
> **Looking forward to your rebuttal feedback!**
>
> Thank you for the feedback! We appreciate you kindly checking our response as the discussion period ends soon. Specifically, we have added new baselines and showed that our proposed RoboTool outperforms the suggested baselines by a large margin. If you have further questions, please do not hesitate to let us know, and we are happy to answer them. Thanks!

---

> ### Author Response · Authors · 2023-11-22
> **Followup on the rebuttal as the discussion period is ending soon**
>
> Dear reviewer b6vg,
>
> With the rebuttal period ending tomorrow, we would like to ask whether our response addressed your questions and alleviated your concerns. If so, could you please kindly consider raising the score?
>
> Best,
>
> Authors

---

### Official Review · Reviewer_QoGZ · 2023-10-31

**Soundness:** 3 good
**Presentation:** 4 excellent
**Contribution:** 3 good
**Rating:** 8
**Confidence:** 3

**Summary:**

This works uses LLMs to generate code that is able to perform some reasoning and planning with a robotic simulated system. It is tested on three different experimental paradigms with two robots. The results provided are impressive. The only drawback of the work is the confusion on what is really planning and control with a tool in the real world and coding a set of skills in a programming environment. Furthermore, the baseline comparison is an ablation study.

**Strengths:**

-	Original solution for planning with reasoning using LLMs.
-	It is able to generate code with a level of reasoning that outperforms previous works.
-	Results are well described and deep.

**Weaknesses:**

-	While the aim proposed by the authors is “we aim to solve a hybrid discrete-continuous planning problem”, this is not solved in this work or at least not described properly.
-	The focus of the paper should be improved. This is a of  language reasoner that generates code. So it is more a programming tool than a RoboTool.
-	Baseline comparison is an ablation study. Thus, the third contribution is not well described.


**Focus**

The clarity of what is achieved should be more clear. The first contribution: “long-horizon hybrid discrete-continuous planning” is not solving the hybrid part. It is using predefined skills. Note that as the authors show just planning is not enough in a real set-up as the world has errors and skills have to be hardcoded. Furthermore, as it is well described in the Limitation this is a very powerful planner that generates code, but the continuous control of the execution is not addressed in this work. In essence this is a very sophisticated planner but it is not solving hierarchical control.

**State of the art**

For completeness I am missing this LLM approach to Robotics: PaLM-E: An Embodied Multimodal Language Model

And also recent works on planning with low-level control such as: Active inference and behavior trees for reactive action planning and execution in robotics. TRO2023

**Results**

For a fair baseline comparison authors can use a PDDL planner as baseline. It may be misleading to call a baseline comparison an ablation study of the own algorithm.

It is not clear the type of randomization in the environment initialization to properly evaluate the accuracy of the planner.

**Questions:**

**Further comments:**

-There is no mention on what type of LLM is being used and how it is pretrained and refined for each component. I think this is important information.

-“Hierarchical Policies for Robot Tool Use” there is no analysis of the combinatorial nature of the parametrized skills. We are talking about 4 skills with how many parameters? How many instances of objects? This is important to understand the level of complexity of the decision tree.

-Do we need 4 LLMs to solve simple reasoning and generate a plan?

*A high-level comment*

Problem solving is the key point in this work. The citation to Josep Call is crucial. While it is shaped as a tool use the fact is that the robot does not understand a tool as a tool but a set of skills that can manipulate the world. Two things that are usually missing in this type of approaches are:

-	Humans we have mechanical/dynamics knowledge learnt from experience. Although the authors mention the affordances, note that it is not only about semantics but also about the real interaction in the environment.

-	There is no analysis of how the system handles uncertainty resolution and trades off exploitation vs exploration (or intrinsic motivation). This is a important concept in creativity. Only reasoning is not enough to induce creativity, but probably reasoning and uncertainty resolution to try new things could be artificial creativity.

---

> ### Author Response · Authors · 2023-11-17
> **Response to Reviewer QoGZ (Part 1)**
>
> Thank you for the encouraging review! We are glad that you appreciate the thoroughness and depth of our work. Here are our responses:
>
> > Focus of "long-horizon hybrid discrete-continuous planning".
>
> Thanks for the question. We explain the hybrid task-motion planning problem here in more detail. An important part of TAMP problems is the interdependence of the motion-level and task-level aspects of the problem. Ignoring such interdependence between them will make it unable to solve the problem. We borrow this example from the seminal TAMP survey [1]:
>  - Consider a task where the robot needs to place a particular object (name $A$) at a location. It might select a high-level plan "skeleton" such as "moveF$(q_0, \tau_1, q_1, p_0)$, pick[$A$]$(q_1, p_0, g)$, moveH[$A$]$(g, q_1, \tau_2, q_2)$, place[$A$]$(q_2, p_1, g)$", where
>      - moveF action involves robot movement when the gripper has no object with it.
>      - moveH[$A$] action involves robot movement when the gripper is holding object $A$.
>  - The action primitives are discrete options. However, the free parameters are continuous values, such as robot configurations $(q_0, q_1, q_2)$, a grasp pose $(g)$, placement poses $(p_0, p_1)$, paths $(\tau_1, \tau_2)$. The skeleton imposes constraints on the choices of those values, or conversely, there could be no satisfying set of values, and the skeleton needs to be changed. For example, if there is object $B$ occupying the target location, the robot needs to first remove $B$ and then place $A$ at the target location.
>
> This example showcased the importance of hybrid discrete-continuous planning. We will modify the problem formulation accordingly in the final version to highlight this.
>
> **Ref:**
>
> [1] Garrett, Caelan Reed, et al. "Integrated task and motion planning." Annual review of control, robotics, and autonomous systems 4 (2021): 265-293.
>
> > Analysis of the combinatorial nature of the parametrized skills.
>
>  - For the robotic arm, we have a skill library that includes```[get_position, get_size, open_gripper, close_gripper, move_to_position```. Among them, ```get_position``` and ```get_size``` take in discrete ```object_name``` as input, while ```move_to_position``` takes in 3D position of the target gripper as input. The longest plan (Can-Grasping) requires about 15 steps. A more detailed description of the parameterized skills can be found in Appendix B in the original draft.
>
> > State of the Art
>
>  - Thanks to the reviewer for pointing out these two papers. We added them into the additional related work in the appendix and will incorperate them to the final version of related work.
>
> > Experiment results
>  - **Compare with a PDDL algorithm**: Although we formulated creative tool use as a planning problem, solving such tasks requires more than just geometric planning. PDDL and other formal-logic planners, for example [2], are able to solve the Can-Grasping task, which requires only geometric planning. However, they cannot solve some other tasks in this work, for example, Cube-Lifting, which requires identifying a hidden lever structure in the scene and finding a way to activate this mechanism. The traditional planners are not able to reason based on common knowledge or manufacture nonexistent tools, which are critical to solving other creative tool-use tasks.
>  - **Randomization:** For robot arm experiments, we randomize the initial configuration of objects. For quadrupedal robot experiments, we randomize the initial position of movable objects and the robots. We show the randomization range in Table 1 and add it to Appendix G in the updated draft.
>
> Table 1: Randomization of objects in each task.
> | | Objects: Randomization range [x_min, x_max, y_min, y_max] |
> | ------------- | ------- |
> | Milk-Reaching | pineapple toy, lock, cube, tomato toy: [0, 0.45, -0.25, 0.25]; hammer: [0, 0.45, -0.25, 0]; milk: [0.45, 0.6, 0, 0.25]  |
> | Can-Grasping | stick: [0, 0.45, -0.25, 0]; paper: [0, 0.45, 0, 0.25]; can: [0.45, 0.7, -0.1, 0.1]  |
> | Button-Pressing | magnetic_cube1, magnetic_cube2, magnetic_cube3: [0, 0.1, -0.25, 0.25]; button: [0.45, 0.7, -0.25, 0.25]  |
> | Sofa-Traversing | surfboard: [0.3, 0.6, -0.175, 0.0]; cloth: [0.3, 0.6, 0.0, 0.2]; robot: [-0.5, -0.2, -0.2, 0.2];   |
> | Sofa-Climbing | box_2: [0.0, 2.0, 0.0, 1.0]; robot: [0.0, 3.0, 0.0, 2.0];   |
> | Cube-Lifting | chair: [0.8, 1.2, -0.1, 0.1]; surfboard: [0.8, 1.2, 0.3, 0.5]; robot: [0.0, 3.0, 0.0, 2.0];   |
>
> **Ref:**
>
> [2] Toussaint, Marc, Kelsey Allen, Kevin Smith, and Joshua Tenenbaum. 2018. “Differentiable Physics and Stable Modes for Tool-Use and Manipulation Planning.” In Robotics: Science and Systems XIV. Robotics: Science and Systems Foundation. https://doi.org/10.15607/rss.2018.xiv.044.

---

> > ### Author Response · Authors · 2023-11-17
> > **Response to Reviewer QoGZ (Part 2)**
> >
> > > Which LLM has been used?
> >
> >  - In the introduction on page 2, we mentioned that "All of these components are constructed using GPT-4." We will highlight it again in the experiment part.
> >
> >
> >
> >
> > > Why 4 LLMs instead of 1 LLM?
> >
> >  - Thanks for the insightful question. When developing RoboTool, we noticed that LLMs are better at handling homogenous tasks rather than heterogeneous ones. Such an observation has also been discovered by prior literature [3]. To get a quantitative comparison, we conducted experiments by concatenating all the prompts of different modules in RoboTool and calling got-4.0 at once. We get a final success rate of 0.03, which is much worse than the proposed RoboTool.
> >
> > **Ref:**
> >
> > [3] Yu, Wenhao, et al. "Language to Rewards for Robotic Skill Synthesis." arXiv preprint arXiv:2306.08647 (2023).
> >
> >
> >
> > > High-level comment.
> >
> >  - Thank you for the insightful comment. We agree with the reviewer about how affordances related to physics properties in robotics. Indeed, physics-grounded knowledge is key to robust interaction with the environment. Recent work has started to investigate this direction, such as physically grounded Vision-Language Models [4]. We also believe that the real-world interactions would enable a more robust embodied LLM system, which would be an interesting future direction for RoboTool.
> >  - In this work, "creative" refers to using tools beyond their designated or default functionality. As discussed above, we think exploration with physical interactions might also lead to more interesting, diverse, and creative solutions. We appreciate the reviewer for the insightful comments.
> >
> > **Ref:**
> >
> > [4] Gao, Jensen, et al. "Physically Grounded Vision-Language Models for Robotic Manipulation." arXiv preprint arXiv:2309.02561 (2023).

---

> > ### Comment · Reviewer_QoGZ · 2023-11-20
> > **Good results, be careful with the terminology.**
> >
> > Thanks so much for the response. After reading the reviews and comments I would keep my score due to the novelty and the good presentation of the results showed. Please be careful with the terminology. For instance in a decision making problem I may say for instance discrete-time, discrete-action, continuous-discrete hybrid state.

---

> > > ### Author Response · Authors · 2023-11-20
> > > **Thank you!**
> > >
> > > Thank you so much for the encouraging review! We will keep polishing the language and improving the manuscripts.

---

### Official Review · Reviewer_pWWx · 2023-11-01

**Soundness:** 2 fair
**Presentation:** 2 fair
**Contribution:** 2 fair
**Rating:** 6
**Confidence:** 3

**Summary:**

The authors propose a prompting approach to enable creative tool use for robots. The approach constsists of four stages, using an "analyzer" prompt to extra objects, a planner planner prompt to generate a rough plan, a "calculator" prompt to populate it with action parameters, and finally a "coder" prompt to generate executable python code. The approach is demonstrated on a simulated and real robotics example.

**Strengths:**

- Getting robots to solving complex tasks is an important and difficult problem
- Many are interested in LLMs at the moment, and this paper provides some further information on how to use them
- The results seem impressive, and it has an ablation study showcasing that each module seems to be required for success on these examples

**Weaknesses:**

- A lot of engineering seems to have gone into these examples. The prompts (on separate github) contain a number of hints on what not to do when solving the problem, which seem engineered for the particular tasks. Examples:
  - "If you do not know the actual value, use an offset = 1m."
  - "You must be careful when calculating with negative values."
  - "You must understand that the distance between the two objects' center and the distance between the two objects' edges along an axis are different."
  - The coder in Fig.2 also generates a seemingly arbitrary gripping offset for the hammer which I guess you engineered depending on the shape of the hammer.
  - Some motion primitives are a bit contrived: As a roboticist, getting the robot to kick the surfboard in place to traverse the sofa seems like an extremely challenging tasks that I guess you just spent a lot of time engineering the motion primitives for. I don't think these are very realistic examples of your approach considering how much engineering must have gone into them. It is very difficult to say how much going on here is just simple symbolic task planning vs. motion planning (e.g. the real-valued positions and orientation parameters).
- It relies only on ChatGPT 4.0 which means that it is unclear to me if this architecture design and ablation study would generalize to other LLMs. GPT4.0 is much better than open source models so the decision is understandable but it is a weakness of the paper. It also makes reproducing it harder since ChatGPT is updated and has been observed to change behavior over time (OTOH it is very easy to use the current version of GPT4...).
- Relation to other LLM works that do manipulation could maybe be clarified, e.g. VoxPose was kind of brushed off as being multi-modal, but isn't that a strength? IIRC they also show somewhat complex actions (grabbing a toast and puttig it on a cutting board). Your example is more complex but it is difficult to quantify since they seem heavily engineered.

Minor issues with presentation/claims:
- Why are the prompts in the appendix only links to github instead of actually in the appendix? IIRC there is no page limit on the ICLR appendix.
- The language could be better, especially the examples in the intro do not actually seem to be grammatically correct calls for action, e.g., "grasping a milk cartoon" should be "grasp a milk cartoon", "walking to the sofa" should be "walk to the sofa" and so on. This is a bit problematic if a prompt is unintentionally ambigious as it could affect your results.

**Questions:**

- How much do you vary/randomize in your sim and real robot examples, does the robot and world always start in the same configuration?
- Can you also say something about how much the action parameters are varied in response to these, or maybe show us some example python code from different runs of the algorithm on e.g. the milk example?

---

> ### Author Response · Authors · 2023-11-17
> **Response to Reviewer pWWx (Part 1)**
>
> Thank you for your constructive feedback. We are pleased to hear that you found our experiments impressive and that the modules in RoboTool contribute to performance improvements. Please find our detailed responses below.
>
> >  The prompts contain hints on what not to do when solving the problem.
>
>  - We removed the prompt related to "offset=1.0" and "careful when calculating negative values." After rerunning the experiments, we found some decrease in RoboTool's success rate (0.87 -> 0.83); however, it was insignificant. We updated Table 1 in the draft accordingly.
>  - The prompts related to the negative values and distances aim to remedy LLMs' deficiency, which hardly hints at how to plan for a specific task. Please note that we use the same prompts for all experiments, not specifically engineered for each task. Designing prompts containing general rules to follow is common in many embodied LLMs literature.
>      - Regarding the prompt mentioning that the distance between objects' centers and boundaries are different, it is related to the 3D reasoning capability of gpt-4: It sometimes mistakenly thinks that the distance between objects is the distance between their centers without considering the size of the objects. We want to reduce such mistakes by reinforcing the distance concept in 3D world. However, RoboTool decides whether to use the distance between centers or the distance between boundaries for downstream planning.
>
> > Is the gripping offset in Fig. 2 code for the hammer engineered based on the shape of the hammer?
>
>  - The numbers are not engineered based on the shape of the hammer. The calculator prompts contain an example of calculating a generic object's grasping position. In other generated codes, RoboTool follows the examples in the prompt and automatically gets the grasping offset based on the object's bounding box size. We provide more generated codes in Appendix I.
>  - We need such an example in the prompt since RoboTool uses general motion primitives, such as ```move_to_position(pos)```  and ```close_gripper()``` to provide extra planning flexibility, instead of more granular primitives, such as ```pick(object)``` and ```place(object)``` in many existing methods [1, 2, 3, 4]. RoboTool needs to calculate an accurate gripper target position, which is difficult to rely solely on the prior knowledge of LLMs alone.
>
> **Ref:**
>
> [1] Singh, Ishika, et al. "Progprompt: Generating situated robot task plans using large language models." 2023 IEEE International Conference on Robotics and Automation (ICRA). IEEE, 2023.
>
> [2] Ahn, Michael, et al. "Do as i can, not as i say: Grounding language in robotic affordances." arXiv preprint arXiv:2204.01691, 2022.
>
> [3] Huang, Wenlong, et al. "Grounded decoding: Guiding text generation with grounded models for robot control." arXiv preprint arXiv:2303.00855, 2023c.
>
> [4] Liang, Jacky, et al. "Code as policies: Language model programs for embodied control." 2023 IEEE International Conference on Robotics and Automation (ICRA). IEEE, 2023.
>
> > The engineering effort put in motion primitives.
>  - Thanks for the question. Our ```push_to_position()``` skill is based on Unitree's built-in walking mode gait with a small gait height and a motion planner to generate waypoints to move objects around (as shown in Appendix B.1 in the original draft). There is surging interest in obtaining better loco-manipulation skills [6-8], which is manipulating objects with a quadrupedal robot. We provide one simple implementation of such loco-manipulation skill, which is already sufficient to perform complex long-horizon tasks. Although obtaining such motion primitive is not the focus of this work, we are eager to integrate more diverse skills in RoboTool.
>  - Instead, in this work, we focus on high-level task planning and assume access to a library of existing motion primitives, similar to SayCan [2], GroundedDecoding [3], PPDL with LLMs [5] etc. Please note that our method is general and is compatible with skills derived from different methods, such as classical planning, RL, or imitation learning.
>
>
> **Ref:**
>
> [5] Silver, Tom, et al. "Generalized planning in pddl domains with pretrained large language models."
> arXiv preprint arXiv:2305.11014, 2023.
>
> [6] Nachum, Ofir, et al. "Multi-agent manipulation via locomotion using hierarchical sim2real." arXiv preprint arXiv:1908.05224 (2019).
>
> [7] A. Rigo, Y. Chen, S. K. Gupta, and Q. Nguyen, “Contact optimization for non-prehensile loco-manipulation via hierarchical model predictive control,” arXiv preprint arXiv:2210.03442, 2022.
>
> [8] M. Sombolestan and Q. Nguyen, “Hierarchical adaptive locomanipulation control for quadruped robots,” arXiv preprint arXiv:2209.13145, 2022.

---

> > ### Author Response · Authors · 2023-11-17
> > **Response to Reviewer pWWx (Part 2)**
> >
> > > Can RoboTool generalize to other LLMs?
> >  - RoboTool is compatible with different LLMs. We provide additional results using both gpt-3.5-turbo and gpt-4. We compare RoboTool with Code as Polices (CaPs) showing the per-task success rates in Table 1 and error breakdowns in Table 2. Table 1 shows that when using different language models, RoboTool could consistently outperform CaPs, showing the generalizability of RoboTool.
> >  - Based on the error breakdowns in Table 2, RoboTool helps reduce the tool use error (whether the correct physical tool is selected) when using gpt-3.5-turbo. However, gpt-3.5-turbo cannot devise a reasonable plan with the selected physical tool. Those observations highlight the tremendous challenges in reasoning, planning, and calculation for creative robot tool use tasks. Such demanding tasks left us with very few options for competent LLMs.
> >  - As suggested by the reviewers, open-sourced LLMs are generally preferable for academic research. However, as in many concurrent works, gpt-4 is found to be unequivocally the best-performing LLM across different tasks and embodiments. We will keep a close look at the open-sourced models and update RoboTool once they achieve performance comparable to gpt-4.
> >
> > Table 1: Additional Baselines.
> > |                 | RoboTool (gpt-4) | RoboTool (gpt-3.5-turbo) | CaPs (gpt-4) | CaPs (gpt-3.5-turbo) |
> > | --------------- | ---------------- | ------------------------ | ---------------------- | ------------------------------ |
> > | Milk-Reaching   | 0.9              | 0.0                      | 0.0                    | 0.0                            |
> > | Can-Grasping    | 0.7              | 0.0                      | 0.2                    | 0.0                            |
> > | Button-Pressing | 0.8              | 0.2                      | 0.6                    | 0.0                            |
> > | Sofa-Traversing | 1.0              | 0.2                      | 0.4                    | 0.0                            |
> > | Sofa-Climbing   | 1.0              | 0.0                      | 0.4                    | 0.0                            |
> > | Cube-LIfting    | 0.6              | 0.0                      | 0.4                    | 0.0                            |
> > | Average         | 0.83             | 0.07                     | 0.33                   | 0.00                            |
> >
> > Table 2: Error breakdowns for new baselines.
> > |     | RoboTool (gpt-4) | RoboTool (gpt-3.5-turbo) | CaPs (gpt-4) | CaPs (gpt-3.5-turbo) |
> > | --- | ---------------- | ------------------------ | ------------ | -------------------- |
> > Tool Use Error| 0.00 | 0.45 | 0.23| 0.96|
> > Logical Error| 0.09 | 0.41 | 0.22 | 0.02|
> > Numerical Error| 0.08| 0.07 | 0.22 | 0.02|
> > Success| 0.83| 0.07 | 0.33| 0.00 |
> >
> > > Comparison with VoxPoser
> >  - Thanks for the question. RoboTool and VoxPoser have different focuses and are complementary. VoxPoser does not focus on complex reasoning or long-horizon task plans like RoboTool. VoxPoser aims to generate low-level trajectories given clear decomposed task instructions. One possible way to integrate the two methods is by using RoboTool to generate the task plan and VoxPoser to generate low-level trajectories.
> >  - Multi-modality is indeed a strength when it comes to LLM agents. Like VoxPoser, RoboTool interacts with [OWL-ViT](https://huggingface.co/docs/transformers/model_doc/owlvit) to detect various objects. As discussed in the limitation section, a promising future direction would be integrating the vision-language model seamlessly into RoboTool. In this paper, we first want to show the initial signs of solving creative tool-use tasks with only LLMs.
> >
> > > Minor issues with the prompt link and task descriptions
> >  - In Appendix D and E of the initial submission, we included hyperlinks that directly jump to the prompts and task descriptions, as we thought that is a cleaner presentation. We sincerely apologize if this choice caused inconvenience to the reviewers. As suggested, we have included the prompts in the updated version instead of links.
> >  - Please note that the descriptions fed to the LLMs contain task descriptions following the template "You want to ...", such as "You want to grasp the milk" or "You want to get on top of the sofa". We apologize for the confusion and correct the descriptions in the introduction as pointed out by the reviewer ("grasping" -> "grasp", "walking" -> "walk").

---

> > > ### Author Response · Authors · 2023-11-17
> > > **Response to Reviewer pWWx (Part 3)**
> > >
> > > > Randomization in sim and real robot examples.
> > >  - Thanks for the question. We indeed randomized the configurations in experiments and sorry for the confusion. For robot arm experiments, we randomize the initial configuration of objects. In quadrupedal robot experiments, we randomize the initial positions of movable objects and the robots. We show the randomization ranges in Table 3 and add it to Appendix G in the updated draft.
> > >
> > > Table 3: Randomization of objects in each task.
> > > | | Objects: Randomization range [x_min, x_max, y_min, y_max] |
> > > | ------------- | ------- |
> > > | Milk-Reaching | pineapple toy, lock, cube, tomato toy: [0, 0.45, -0.25, 0.25]; hammer: [0, 0.45, -0.25, 0]; milk: [0.45, 0.6, 0, 0.25]  |
> > > | Can-Grasping | stick: [0, 0.45, -0.25, 0]; paper: [0, 0.45, 0, 0.25]; can: [0.45, 0.7, -0.1, 0.1]  |
> > > | Button-Pressing | magnetic_cube1, magnetic_cube2, magnetic_cube3: [0, 0.1, -0.25, 0.25]; button: [0.45, 0.7, -0.25, 0.25]  |
> > > | Sofa-Traversing | surfboard: [0.3, 0.6, -0.175, 0.0]; cloth: [0.3, 0.6, 0.0, 0.2]; robot: [-0.5, -0.2, -0.2, 0.2];   |
> > > | Sofa-Climbing | box_2: [0.0, 2.0, 0.0, 1.0]; robot: [0.0, 3.0, 0.0, 2.0]  |
> > > | Cube-Lifting | chair: [0.8, 1.2, -0.1, 0.1]; surfboard: [0.8, 1.2, 0.3, 0.5]; robot: [0.0, 3.0, 0.0, 2.0]  |
> > >
> > >
> > > > How much are the action parameters varied to the randomization? Maybe show us some example Python codes.
> > >
> > >  - Please note that RoboTool usually gets the 3D positions of objects in the scene by invoking the ```get_position(object)``` primitive, and assigning it to a variable in the code for downstream computation with numpy. Therefore, the code may look very similar to each other, but the actual parameter values are different. Note that in Section 5.5, Discriminative Tool-use Capability, we tested RoboTool's ability to use tools when necessary and ignore tools when the robot can directly finish tasks without the need to manipulate other objects. That is when the plan and code look very different when given different task configurations.
> > >  -  We provide more Python codes in Appendix I to show that RoboTool can vary parameters based on the environment.

---

> ### Author Response · Authors · 2023-11-20
> **Looking forward to your rebuttal feedback!**
>
> Thank you so much for your feedback and suggestions to improve our work! We hope that our replies and additional results have addressed your concerns. As the discussion period ends in a few days, we would like to ask if you have any further questions, and we are happy to provide more clarification. Thank you!

---

> ### Comment · Reviewer_pWWx · 2023-11-20
> **Reviwer response to rebuttal**
>
> I thank the authors for their thorough rebuttal and new experiments.
>
> Having gone over this again now, I think the paper has merit but I still have a few concerns w.r.t. robotics framing and the experiments:
> 1) Your first contribution seems to have merit but is a bit arbitrarily narrow. LLMs encode common-sense knowledge about the world, so that you can use them to improve robot interaction with the world is not that surprising in itself. The way you do this using multiple LLM queries is the novel part and seems to have merit based on the ablations you have shown. However this has some weaknesses:
>
>     a) The paper focuses on tool use (probably because of the high-profile role it has played in theories of animal intelligence), but since the embodiment of tool use is abstracted away (relies on pre-engineered perception and motion primitive modules) it seems these methods could be used for any task planning problem. This is a good thing in a sense but it makes the focus on tool use seem like an arbitrary restriction chosen more for PR than practical considerations.
>
>     b) In particular, the paper claims that it solves a motion planning problem but the code examples I've seen seem to mainly only move_to(get_position_of_object(...)) with possible +object_size/2 added or something similar. It is very difficult to get a sense for how much of a motion vs. task planning problem it is actually solving. This is extra problematic because the "calculator" component in your novel architecture is supposed to come up with the (real-valued) parameters but it mainly seems to just need to plug in getters for object positions. The lack of non-trivial motion planning in the experiments also undermines your proposed architecture a bit. These planner properties could probably also have been tested individually instead of everything engineered together into a complex experiment.
>
> 2) Your second contribution on creating a benchmark for tool use is also a bit weak because some of your benchmarks (especially the quadruped one) seem a) a bit contrived and/or b) a lot of work to reproduce. Again, all the hard embodiment problems of tool use are engineered away, and doing this engineering to get a quadruped robot to reliably push objects on sofas seems like potentially a lot of effort. Is there a sim version of this btw?
>
> That said, since the architecture does show improved results in your experiments (even if it is arguably mostly "task planning for tool use", and only evaluates LLM generalization on two different versions of ChatGPT). I will tentatively raise the score one step while I digest the other reviews / discussion.

---

> > ### Author Response · Authors · 2023-11-20
> > **Thank you for the response!**
> >
> > Thank you for spending time and effort on our paper and raising the score! We hope to address your remaining concerns here:
> >
> > > 1a. The focus of creative tool use.
> >
> > The uniqueness of tool use versus other traditional task planning is manifested by making "impossible" tasks possible. A RoboTool agent needs to identify whether a tool is needed, what can be used as a tool, and how to use the tools. The reason why the action primitives are general instead of tool-specific is that we do not want to inject too much inductive bias about how to use them. Tool use stands as one of the most cognitively challenging domains beyond commonly seen manipulation tasks.
> >
> > As the reviewer pointed out, RoboTool can be used for traditional task planning in principle. However, the converse is not necessarily true, as evidenced by our experiments: the LLM agent designed for traditional task planning may not be able to use tools as it requires much more cognition and planning efforts.
> >
> > Not surprisingly, the research community also treats tool use as one of the important topics of LLM agents, although in a different domain (virtual interactions with web browsers, databases, code interpreters, etc) [1]. We think RoboTool is one of the first steps towards using LLMs for creative physical tool use and will inspire more future research.
> >
> >
> > > 1b. Motion planning part seems to be invoking ```pos = get_position(object)``` and then ```move_to_pos(pos)```. Trivial motion planning.
> >
> > We observe that the "calculator" does more than just move to a 3D target position obtained by ```get_position(object)```. Here are a few examples:
> >  - In Sofa-Traversing, when trying to calculate the target position for the surfboard, the calculator outputs ```surfboard_target_position = [(sofa_1_position[0] + sofa_2_position[0]) / 2, ...]```, meaning RoboTool holds the concept of 'center between objects (sofa_1 and sofa_2)' and potentially the suitable target position of the surfboard.
> >  - In Can-Grasping, when trying to drag the strip of paper in the negative x direction, the calculator outputs ```move_to_position(np.array([0.0, paper_position[1], paper_position[2]]))```, meaning RoboTool knows to drag the paper as into the workspace as possible without reaching outside of the workspace ($x \geq 0.0$).
> >  - In Button-Pressing, when trying to press the button, the calculator could output the target position as ```button_position - np.array([magnetic_cube1_size[0] + magnetic_cube2_size[0] + magnetic_cube3_size[0], ... ```, meaning RoboTool understands the concept of the 'cube chain' or 'stick', and subsequently subtracts the length of the chain by summing up the size of three cubes.
> >
> > The examples above show that the calculator does more than just ```get_position(object)``` and indeed plans reactively according to the plan skeleton. Furthermore, please note that we have more fine-grained motion planning, such as RRT, a path planning algorithm built-in in the ```move_to_position()``` skill for collision avoidance (Appendix B.1).
> >
> >
> >
> > > 2. Benchmark seems a bit contrived (especially the quadruped one) and difficult to reproduce without simulators.
> >
> > Indeed, some of our tasks are not commonly seen in daily life since there are no large-scale deployments of embodied agents currently. At the current stage, we view our benchmark more as physical puzzles rather than daily tasks, challenging the limits of SOTA LLMs. With the increasing deployment of robots in real life, we believe there will be more use cases of tool use, as we humans use tools explicitly or implicitly every day.
> >
> > Again, our ```push_to_position()``` skill is based on Unitree's built-in walking mode gait with a small gait height and a motion planner to generate waypoints to move objects around (Appendix B.1). We do have simulators for our benchmark (the simulated success rate is shown in Table 1) built on top of Robosuite. We will release the simulators and motion planners for real robot experiments, together with the RoboTool framework, for reproducibility and benefit to the community.
> >
> > We hope our responses help address your concerns. Please let us know if there is more clarification we can provide while waiting for discussions with other reviewers.
> >
> >
> > **Ref:**
> >
> > [1] Qin, Yujia, et al. "Tool learning with foundation models." arXiv preprint arXiv:2304.08354 (2023).

---

### Author Response · Authors · 2023-11-20
**General Response**

Dear reviewers:

Thank you all for the constructive feedback and encouraging review. We appreciate that the reviewers find that the creative physical tool-use problem we study is "important and difficult" (pWWx); our creative tool-use benchmark is "well-designed as a starting point" (xccj); our method is "original" (QoGz), of broad interest (pWWx), "highly desirable" (b6vg), "effective and compelling" (xccj); and our experiment results are "impressive" (pWWx), "well described and deep" (QoGz).

In addition to the detailed comments to each review, we would like to highlight our contributions and main modifications during the rebuttal phase.

 - **Contributions of this paper (b6vg):**
     - **Novel Problem:** Creative tool use is a significant but underexplored problem in the existing literature. Enhancing robots with this ability could dramatically improve their performance in tasks that were previously unachievable.
     - **Novel Benchmark:** We develope a unique and challenging benchmark specifically designed to evaluate the creative tool use capabilities in robotics. This benchmark includes two types of robot embodiments and three distinct categories of creative tool use. Such a benchmark is essential and non-trivial to design.
     - **Novel Method:** While the planner-coder LLM agents have been previously discussed in the literature, merely applying existing API-based methods like Code as Policies or ViperGPT does not adequately meet the unique cognitive, reasoning, and planning challenges presented by creative tool use. Our results show that baselines were unable to complete the tasks in our benchmark with a high rate of success.
 - **Experiments with New Baselines (pWWx, b6vg, xccj):** We compare RoboTool with Code as Policies and ViperGPT. The results are added to Appendix F. We show that RoboTool significantly outperforms baselines with a considerably higher success rate.
 - **Experiments with Different LLMs (pWWx):** In addition to results using gpt-4 in the initial submission, we use gpt-3.5-turbo as the backbone for each module of RoboTool. We find that RoboTool still demonstrates enhanced performance over Code as Policies when using gpt-3.5-turbo for both methods.
 - **Prompt Cleanp-up (pWWx, xccj):** We remove unnecessary prompts related to offsets and negative values, and include the same amount of information for different objects in each task. Moreover, we add more objects in the Milk-Reaching task and show that RoboTool can still select the hammer among five objects. We provide the updated prompts and task descriptions in Appendix D and E. After prompt cleanup, RoboTool has a slight performance drop, but not significant, as shown in Table 1 in the updated draft.
 - **Randomization Details: (pWWx, QoGZ, xccj):** We add a table showing randomization ranges of objects and robot initial configurations to Appendix G.
 - **Related Work (QoGZ, b6vg):** We summarize the additional related work in Appendix H.

We hope our response and additional experiment results could help address your concerns. Thank you again for your invaluable comments and constructive feedback.

Please do not hesitate to let us know if there are other clarifications we can offer.

Best,
Authors

---

### Meta-Review · Area_Chair_YvBG · 2023-12-05

**Metareview:**

**Summary**: This paper proposes RoboTool, an LLM-guided method for long-horizon planning in tool use. The framework consists of 4 components:  1. an Analyzer that interprets natural language to pick salient task features; 2. a Planner that generates robot plans based on the language and the salient task features; 3. a Calculator that computes skill parameters; 4. a Coder that translates these plans into code. The paper also presents a 6-task benchmark for evaluating tool use. The approach is demonstrated in simulated and real robot experiments, on two robotic platforms: a robot arm and a quadruped.

**Strengths**:
- The paper is well written and clear.
- The evaluation is carried out on two robotic platforms, which is impressive.
- The 6 "physical puzzles" that were devised are very fun. The idea of a physical puzzle benchmark in itself is very compelling.

**Weaknesses**:
- The way they are currently described, the contributions are a bit misleading:
1. As several other reviewers pointed out, “long-horizon hybrid discrete-continuous planning” is misleading because the algorithm is not solving the hybrid part. From the discussion, though, it sounds like the authors plan to change the language here.
2. To echo a few of the reviewers' concerns, is a set of 6 very specific tasks worthy of being called a benchmark? On the one hand, I like the idea of treating these tasks as physical puzzles that push our robots' limits for what they can do; on the other hand, it's only 6 tasks with very specific objects. Again, I love the idea of creating a benchmark with physical puzzles like in this paper, but I think a compelling benchmark would have more than 6 tasks and each task would be more general than "milk-reaching" -- for instance, "milk-reaching" should really just be "far-object-reaching, i.e. each task should be applicable to more than a single set of objects. In its current form, I don't see the 6-task benchmark as a worthy enough contribution, despite the fact that I find the *idea* of a physical puzzle benchmark extremely compelling; I would have *loved* to see a comprehensive benchmark like this, but unfortunately I don't think this paper quite does it justice for the reasons I just described.
3. As other reviewers pointed out, there aren't really any baselines; as far as I can tell, evaluation is an ablation study. There are many baselines that several reviewers brought up, and I believe the authors should test against them.
- There seems to be a *lot* of engineering (of many kinds) to get this pipeline to work. First, there is a lot of prompt engineering (as some reviewers also pointed out by looking at the prompt examples). Second, the chosen tasks are very specific with very specific objects, object positions, and layouts. This calls into question the generality of the proposed approach. One way to alleviate this in the future: For 1:  I think this may have already been addressed in the conversation with the first reviewer, but there are some very specific details in the prompts that give away hints at what the LLM should do and might bias it; please remove them and try to make the prompts as "fair" and unbiased as possible. For 2: vary the object, vary the positions, vary the layouts; sometimes some layouts won't be possible but I think that would be great if the LLM could figure it out.
- I don't think "creative tool use" is the best way to motivate or "brand" this problem. If I understand the paper correctly, the problem presented is that of long-horizon tasks with potentially sparse rewards, not unlike what we'd find in Minecraft or Montezuma's Revenge. For instance, "walk to the other sofa" when there is a gap requires the robot to figure out that it needs to close the gap with another object before it can cross; that's similar to how in Montezuma's revenge the agent needs to figure out that it needs to grab the key before it can open the door, or how in Minecraft you need to chop wood before you can build a house. From this perspective, it is a bit odd that the authors chose to motivate the paper as tackling a novel problem, rather than embrace this already existing and very complex problem.
- The use of LLMs to do long-horizon planning and propose code is not novel. The paper rebrands the 4 LLM components but they all separately exist and have been used in similar ways. Perhaps the use of all components together in this configuration is novel, but the paper gives the reader the impression that each component is novel when it is not. I suggest revising the writing to clarify what is novel vs what exists and is merely being incorporated in a novel framework.

Overall, I think this paper has potential and presents some really interesting (and fun!) physical puzzle tasks. However, I think to be compelling it needs more work on expanding the 6 tasks to more objects and layouts, and toning down the writing to be more accurate and precise about what are the true contributions.

**Justification For Why Not Higher Score:**

I think the 6 tasks are very specific, there are only ablation comparisons no baselines, and the language in the paper oversells the contributions. I am happy to revise my score to an accept as poster though upon further discussion with the SAC.

**Justification For Why Not Lower Score:**

N/A

---

### Decision · Program_Chairs · 2024-01-16

Reject